# MEAL: A Benchmark for Continual Multi-Agent Reinforcement Learning

**Tristan Tomilin**[1]  **Luka van den Boogaard**[1]  **Samuel Garcin**[2]  **Constantin Ruhdorfer**[3]  **Bram Grooten**[1]
**Fabrice Kusters**[1]  **Yali Du**[4]  **Andreas Bulling**[3]  **Mykola Pechenizkiy**[1]  **Meng Fang**[5][1]

## Abstract

Benchmarks play a central role in reinforcement learning (RL) research, yet their computational constraints often shape what is studied. Despite the motivation of lifelong learning, most continual RL papers consider only 3–10 sequential tasks, as CPU-bound environments make longer sequences impractical. Meanwhile, continual learning (CL) in cooperative multi-agent settings remains largely unexplored. To address these gaps, we introduce **MEAL** (**M**ulti-agent **E**nvironments for **A**daptive **L**earning), the first benchmark for continual multi-agent RL. By leveraging JAX and GPU acceleration, MEAL enables training on sequences of 100 tasks on a single GPU in a few hours. We find that long task sequences reveal failure modes that do not appear at smaller scales.

## 1. Introduction

Continual RL has recently attracted growing interest (Hafez & Erekmen, 2024; Chen et al., 2024; Chung et al., 2024; Erden et al.), but progress in the field remains constrained by computational limitations. CRL inherently requires sequential training combined with repeated evaluation on previously encountered tasks, causing both training and evaluation time to scale with sequence length. As a result, most existing benchmarks are restricted to short task sequences, typically on the order of 5-15 tasks (Sorokin & Burtsev, 2019; Powers et al., 2022; Tomilin et al., 2023; Liu et al., 2025), limiting insight into long-horizon learning dynamics.

These limitations become even more pronounced in multi-agent settings. Although continual multi-agent reinforcement learning (CMARL) introduces unique challenges, it remains largely unexplored (Yuan et al., 2023; 2024). In cooperative environments, agents must establish implicit conventions or roles to coordinate effectively (Strouse et al., 2021). As tasks or environment dynamics change, such conventions may break down, and inter-agent dependencies can amplify individual forgetting into team-level failures. Unlike traditional MARL, CMARL introduces non-stationarity through an evolving task distribution and changing cooperation partners (Yuan et al., 2024). Studying these effects at scale requires benchmarks that support long task sequences and efficient evaluation, which current CPU-bound environments with limited tasks struggle to provide.

To address these gaps, we introduce **MEAL**[1], the first benchmark for continual MARL built around an end-to-end JAX-based pipeline. JAX's just-in-time compilation, vectorization, and GPU execution enable high-throughput simulation and training, making 100-task sequences feasible on a single GPU within a few hours. MEAL enables the study of forgetting, transfer, network plasticity, curriculum learning, and partner adaptation in cooperative multi-agent settings.

The **contributions** of our work are three-fold. (1) We introduce MEAL, the first benchmark for CMARL, and the first CRL benchmark with hardware-accelerated execution, scaling training on modest hardware to task sequences far longer than existing benchmarks allow. (2) To facilitate very long task sequences, we provide procedural task generation with scalable difficulty across four cooperative JAX environments. (3) We evaluate CL methods combined with MARL algorithms, showing that experimental conclusions can change as task sequences lengthen, and that retaining cooperative behavior is a distinct challenge that worsens with more agents, harder tasks, and changing partners.

## 2. Related Work

**Continual Reinforcement Learning.** CRL studies how agents learn tasks sequentially without forgetting previous knowledge. Popular methods include regularization-based approaches such as EWC (Kirkpatrick et al., 2017), architectural strategies like PackNet (Mallya & Lazebnik, 2018); and replay-based methods such as AGEM (Chaudhry et al., 2018). However, the behavior of these methods under multi-agent coordination remains largely unexplored.

[1]Eindhoven University of Technology, The Netherlands [2]University of Edinburgh, UK [3]University of Stuttgart, Germany [4]King's College London, UK [5]University of Liverpool, UK. Correspondence to: Tristan Tomilin <t.tomilin@tue.nl>, Meng Fang <meng.fang@liverpool.ac.uk>.

*Proceedings of the 43$^{rd}$ International Conference on Machine Learning*, Seoul, South Korea. PMLR 306, 2026. Copyright 2026 by the author(s).

---

[1]The code and environments are accessible on GitHub.

*Table 1.* **Comparison of CRL and MARL benchmarks**. MEAL combines continual learning and multi-agent reinforcement learning, unifying properties that are typically studied in isolation. It is the first to enable GPU-accelerated continual RL. Procedurally generated environments with scalable difficulty ensure that the benchmark remains relevant as methods improve.

| Benchmark | PCG | Scalable Difficulty | GPU-accelerated | Multi-Agent | Continual Learning | Action Space | Reference |
|---|---|---|---|---|---|---|---|
| CORA | ✓/ ✗ | ✗ | ✗ | ✗ | ✓ | Mixed | (Powers et al., 2022) |
| MPE | ✗ | ✗ | ✗ | ✓ | ✗ | Continuous | (Mordatch & Abbeel, 2018) |
| SMAC | ✗ | ✓ | ✗ | ✓ | ✗ | Discrete | (Samvelyan et al., 2019) |
| Continual World | ✗ | ✗ | ✗ | ✗ | ✓ | Continuous | (Wołczyk et al., 2021) |
| Melting Pot | ✗ | ✗ | ✗ | ✓ | ✗ | Discrete | (Agapiou et al., 2022) |
| Google Football | ✗ | ✓ | ✓ | ✓ | ✗ | Discrete | (Kurach et al., 2020) |
| JaxMARL | ✓/ ✗ | ✗ | ✓ | ✓ | ✗ | Mixed | (Rutherford et al., 2024b) |
| COOM | ✗ | ✗ | ✗ | ✗ | ✓ | Discrete | (Tomilin et al., 2023) |
| SocialJax | ✗ | ✗ | ✓ | ✓ | ✗ | Discrete | (Guo et al., 2025) |
| Continual Bench | ✗ | ✗ | ✗ | ✗ | ✓ | Continuous | (Liu et al., 2025) |
| **MEAL** | ✓ | ✓ | ✓ | ✓ | ✓ | Mixed | |

**Multi-Agent Reinforcement Learning.** In MARL, multiple agents learn to act in a shared environment, often under partial observability and either cooperative or competitive goals (Hernandez-Leal et al., 2019; OroojlooyJadid & Hajinezhad, 2019). In cooperative MARL, agents share a common reward signal and must learn to coordinate their behaviors, making credit assignment and non-stationarity central challenges (Lowe et al., 2017; Foerster et al., 2018).

**Continual MARL.** Lifelong Hanabi (Nekoei et al., 2021) introduces a testbed for evaluating whether agents can coordinate with unseen teammates. MACPro (Yuan et al., 2024) uses learned task contextualization and progressive multi-head expansion to handle evolving tasks. RPG (Yao et al., 2025) learns task-agnostic relational representations and uses a conditional hypernetwork to generate task-specific policies, enabling continual adaptation in MARL. CO-MAD (Xiao et al., 2026) addresses CMARL in the offline setting, discovering reusable coordination skills from offline data and continually expanding a skill library across a task stream to counter interference and forgetting.

**Benchmarks.** Popular MARL benchmarks include SMAC (Samvelyan et al., 2019), MPE (Mordatch & Abbeel, 2018), Google Football (Kurach et al., 2020), and Melting Pot (Agapiou et al., 2022). While widely used to study coordination and multi-agent interaction, they are not designed for continual learning.

Rather than designing entirely new simulators from scratch, many CRL benchmarks are constructed by organizing existing RL environments into task sequences. Leveraging platforms that are already well understood by the community lowers the barrier for entry. COOM (Tomilin et al., 2023) reuses and designs new tasks in ViZDoom (Kempka et al., 2016). CORA (Powers et al., 2022) composes tasks from Atari, Procgen, MiniHack and CHORES. Continual Bench (Liu et al., 2025) reuses Meta-World (Yu et al., 2020) task primitives, enforcing a unified world dynamics,

whereas Continual World (Wołczyk et al., 2021) directly sequences Meta-World tasks. Despite reusing an existing environment suite, Continual World has become one of the most widely adopted CRL benchmarks (Pan et al., 2025).

In parallel, the RL benchmarking community has seen a strong shift toward JAX-based simulation environments, driven by the need for efficient vectorization and hardware acceleration. Recent benchmarks and platforms such as JaxMARL (Rutherford et al., 2024b), Social-JAX (Guo et al., 2025), Craftax (Matthews et al., 2024a), and Kinetix (Matthews et al., 2024b) demonstrate the advantages of JAX for large-scale and accelerated experimentation. However, to the best of our knowledge, no CRL benchmark has adopted JAX as its underlying framework.

MEAL follows these two established benchmarking paradigms by building on JaxMARL, addressing continual MARL at scale. By adopting a fully JAX-based implementation, MEAL is the first CRL benchmark to enable hardware-accelerated, large-scale evaluation, making long-horizon continual experiments that were previously computationally infeasible practical.

## 3. Preliminaries

**Cooperative Multi-Agent MDP** We consider a fully observable cooperative Markov game $\langle N, S, A, P, R, \gamma \rangle$, with $N$ agents, state space $S$, joint action space $A = A^1 \times \cdots \times A^N$, transition function $P : S \times A \times S \to [0, 1]$, shared reward function $R : S \times A \times S \to \mathbb{R}$, and discount factor $\gamma \in [0, 1)$. Each agent observes the full state $s \in S$ at every time step.

**Continual MARL** We consider a continual MARL setting in which a shared policy $\pi_\theta = \pi_{\theta}^i{}_{i \in N}$ is learned over a sequence of tasks $\mathcal{T} = \mathcal{M}_1, \ldots, \mathcal{M}_N$, where each $\mathcal{M}_i = \langle N, S_i, A, P_i, R_i, \gamma \rangle$ is a fully observable cooperative Markov game with consistent action and observation spaces. At training phase $i$, agents interact exclusively with

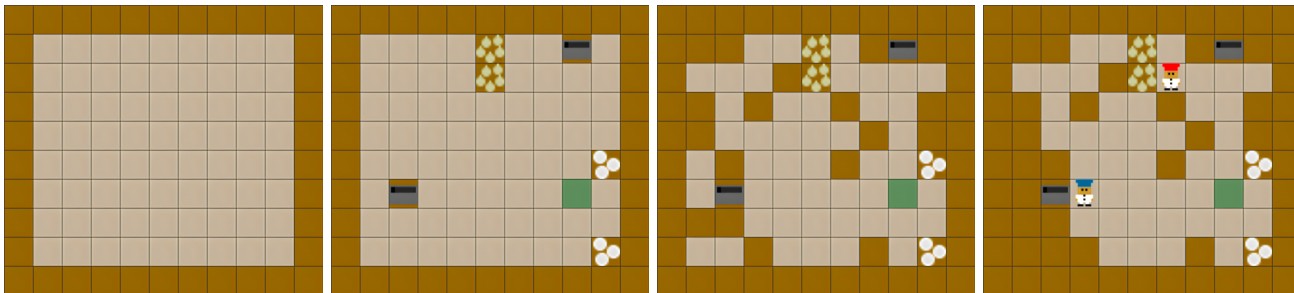

*(a)* Empty grid drawn with outer walls.   *(b)* Interactive stations sampled at random locations.   *(c)* Grid filled with walls to match obstacle density.   *(d)* Agents added and unreachable tiles pruned.

*Figure 1.* Procedural generation pipeline of a **Level 3** layout. Starting from an empty grid with outer walls, the generator injects interactive stations, adds walls to match the desired obstacle density, places agents, and finally prunes unreachable tiles.

$\mathcal{M}_i$ for a fixed number of iterations $\Delta$, collecting trajectories $\tau_{i,1}, \ldots, \tau_{i,\Delta}$ to update their policy. Past tasks are inaccessible, and no joint training is allowed. The objective is to maximize performance on all tasks in the sequence.

## 4. MEAL

We present MEAL, the first continual MARL benchmark, built on the JaxMARL (Rutherford et al., 2024b) version of Overcooked (Carroll et al., 2019), a widely used cooperative MARL environment (Hu et al., 2020; Wu et al., 2021; Strouse et al., 2021), providing a familiar and well-studied foundation. The goal in Overcooked is for agents to cooperatively prepare and deliver soup in a grid-based kitchen. They must collect onions, place them into pots, wait for the soup to cook, plate the dish, and deliver it to a serving station. Further details about the environment are provided in Appendix B. Overcooked poses credit assignment challenges and the need for precise coordination, as agents must execute tightly coupled action sequences (Hernandez-Leal et al., 2019). Prior work has shown that agents often overfit to spurious correlations in fixed layouts, resulting in poor generalization under minor changes (Knott et al., 2021). This makes Overcooked particularly well-suited for CL, as even small layout variations can induce substantial distributional shifts. Although MEAL is not tied to a single domain, we deliberately center this work on Overcooked to enable deeper, more focused analysis rather than spreading evaluation thinly across many disparate environments. To demonstrate that MEAL extends beyond Overcooked, we incorporate JAXNAV (Rutherford et al., 2024a), SMAX (Rutherford et al., 2024b), and MPE SIMPLESPREAD (Lowe et al., 2017; Mordatch & Abbeel, 2018) (see Appendix O). Built entirely in JAX (Bradbury et al., 2018), MEAL is the first CRL benchmark with hardware-accelerated execution. Its limitations are discussed in Appendix P.

### 4.1. MEAL Generator

Existing CRL benchmarks for the task-incremental setting provide a limited set of tasks (Sorokin & Burtsev, 2019;

Powers et al., 2022; Tomilin et al., 2023). To facilitate long sequences, we procedurally generate new Overcooked kitchens on the fly. The generator $G$ draws a random width and height from the specified range, places an outer wall, then sequentially injects the interactive tiles (goal, pot, onion pile, plate pile), extra internal walls to match the target obstacle density, and finally, the agents' starting positions. Figure 1 depicts the steps in this pipeline, and the process is described more in-depth in Appendix A.2. Each candidate grid is accepted only if a built-in validation module confirms that both agents can complete at least one cook–deliver cycle. This yields a continuous space of solvable, variable-sized kitchens that we can learn continually. Appendix A.3 brings further details about the validator. This approach offers a virtually infinite supply of layouts. For reproducibility, the generation process can be fully controlled via a user-specified seed.

### 4.2. Layout Difficulty

To enable meaningful comparison and analysis of existing methods while also providing headroom for more capable future methods, we design a scalable difficulty system, varying (1) grid width, (2) grid height, and (3) obstacle density. This approach yields diverse configurations while maintaining consistent complexity within each level. Figure 2 depicts layouts of each level. As the grid size and wall tile coverage increase, agents must develop more sophisticated coordination strategies. Higher difficulty layouts feature longer paths between key items, tighter bottlenecks, and greater structural variability, all of which make exploration, retention, and adaptation more challenging. While in this work we consider three difficulty levels, it is trivial to extend it further. Although obstacle density has a practical upper bound, environment complexity can be scaled arbitrarily by increasing grid size without altering task semantics.

### 4.3. Continual Learning Sequences

Instead of a continuous domain shift, MEAL provides discrete task sequences $\mathcal{T} = (\mathcal{M}_1, \ldots, \mathcal{M}_N)$ with boundaries.

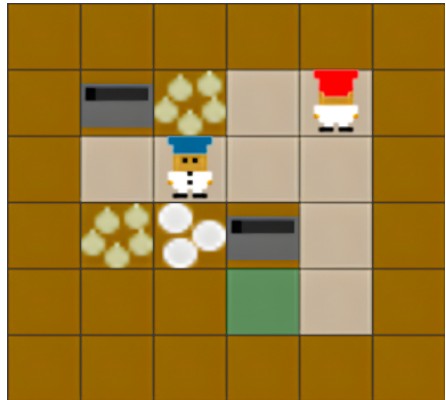 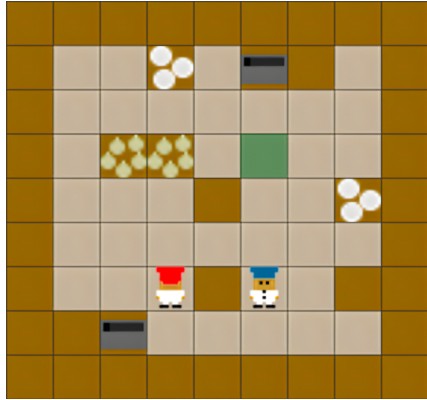 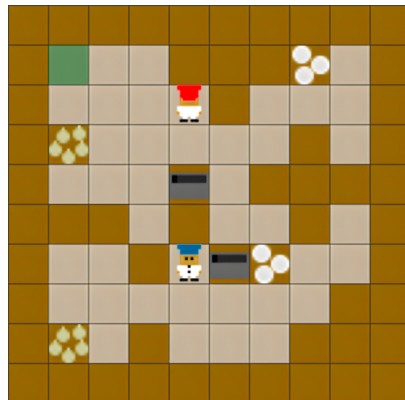

*(a)* **Level 1** (Easy): $6 \leq$ width/height $\leq 7$, obstacle density $\approx 15\%$. Layouts are compact, making exploration easy. Interactable items are close together, making travel distances short. Agents can often complete the task independently with no coordination.

*(b)* **Level 2** (Medium): $8 \leq$ width/height $\leq 9$, obstacle density $\approx 25\%$. Exploration is harder as stations are more spread out. Layouts often have chokepoints, requiring agents to coordinate movement and avoid blocking one another.

*(c)* **Level 3** (Hard): $10 \leq$ width/height $\leq 11$, obstacle density $\approx 35\%$. The map is likely split into disjoint regions, making it impossible for one agent to cook the soup alone. Agents must pass ingredients and dishes across counters and divide the tasks.

*Figure 2.* Overcooked layouts generated at each difficulty level. Increasing grid size and obstacle density lead to longer travel distances, harder exploration, and greater coordination demands.

**Kitchen Layouts** Following most CRL research, which studies non-stationarity from changing environments, our primary task sequences vary the kitchen layout. Given the difficulty-level generators $G_\ell$ and a fixed seed, we explore three sequence regimes: (i) **fixed-level**, where all $N$ tasks are sampled i.i.d. from a single $G_\ell$; (ii) **curriculum**, where equal numbers of tasks are drawn from $G_1$, $G_2$, $G_3$ in ascending order (Appendix F); and (iii) **repetition**, where a base sequence is repeated $r$ times to study plasticity loss (Section 5.5).

**Diverse Partners** Cooperative MARL exposes a second source of non-stationarity that single-agent CL cannot capture, since an agent's success depends not only on the environment but also on its partners. Ad-hoc teamwork (AHT, Stone et al., 2010) studies coordination with partners whose strategies are unknown beforehand, but usually frames this as a zero-shot problem against a single fixed partner. AHT methods are typically evaluated against diverse partner populations as a proxy for human-AI coordination and robustness to varied strategies (Strouse et al., 2021; Zhao et al., 2023; Yan et al., 2023; Wang et al., 2025; Ruhdorfer et al., 2025b). Lifelong Hanabi (Nekoei et al., 2021) already casts ad-hoc teamwork as CL for a card game. MEAL brings this into spatial cooperative environments within a unified, GPU-accelerated pipeline. The agent adapts to a sequence of changing partners while retaining the ability to coordinate with earlier ones. Following prior work (Wang et al., 2025; Ruhdorfer et al., 2025a), we generate diverse partners by combining (i) hardcoded strategies (random, static), (ii) planning-based agents (onion-only, plate-only, and a human-like planner with stochastic task selection), and (iii) populations trained with best-response diversity (BRDiv,

Rahman et al., 2023), which maximizes self-play performance while minimizing cross-play compatibility.

### 4.4. Evaluation Metrics

We measure task performance by the number of soups delivered per episode. Since MEAL layouts vary greatly in size, structure, number of interactive stations, and distances between them, raw delivery counts are not directly comparable. We therefore normalize the soup delivery by the optimal cook-deliver cycle for a single agent on any given task (see Appendix A.1). We account for the cooking time, pickup/drop interactions, shortest paths between onion piles, pots, plate piles, and delivery counters. A normalized soup delivery of 1 indicates that the agent(s) achieved the optimal single-agent performance, while values above 1 reflect effective cooperation that exceeds solo efficiency. Let $s_i(t)$ denote this normalized soup delivery on task $i$ at timestep $t$. A training sequence of $N$ tasks, each lasting $\Delta$ steps, results in a total of $T = N \cdot \Delta$ timesteps. The $i$-th task is therefore trained during the interval $t \in [(i-1)\Delta, i\Delta]$. Following prior work on CRL (Wołczyk et al., 2021; Tomilin et al., 2023), we rely on three core metrics to study CRL in the multi-agent setting.

**Average Soup Delivery** As our core performance metric, we report the mean soup delivery across all tasks at the end of training, reflecting the overall balance between stability and plasticity:

$$\mathcal{S} = \frac{1}{N} \sum_{i=1}^{N} s_i(T). \tag{1}$$

**Forgetting** Forgetting quantifies the decline in performance on past tasks due to interference from training on

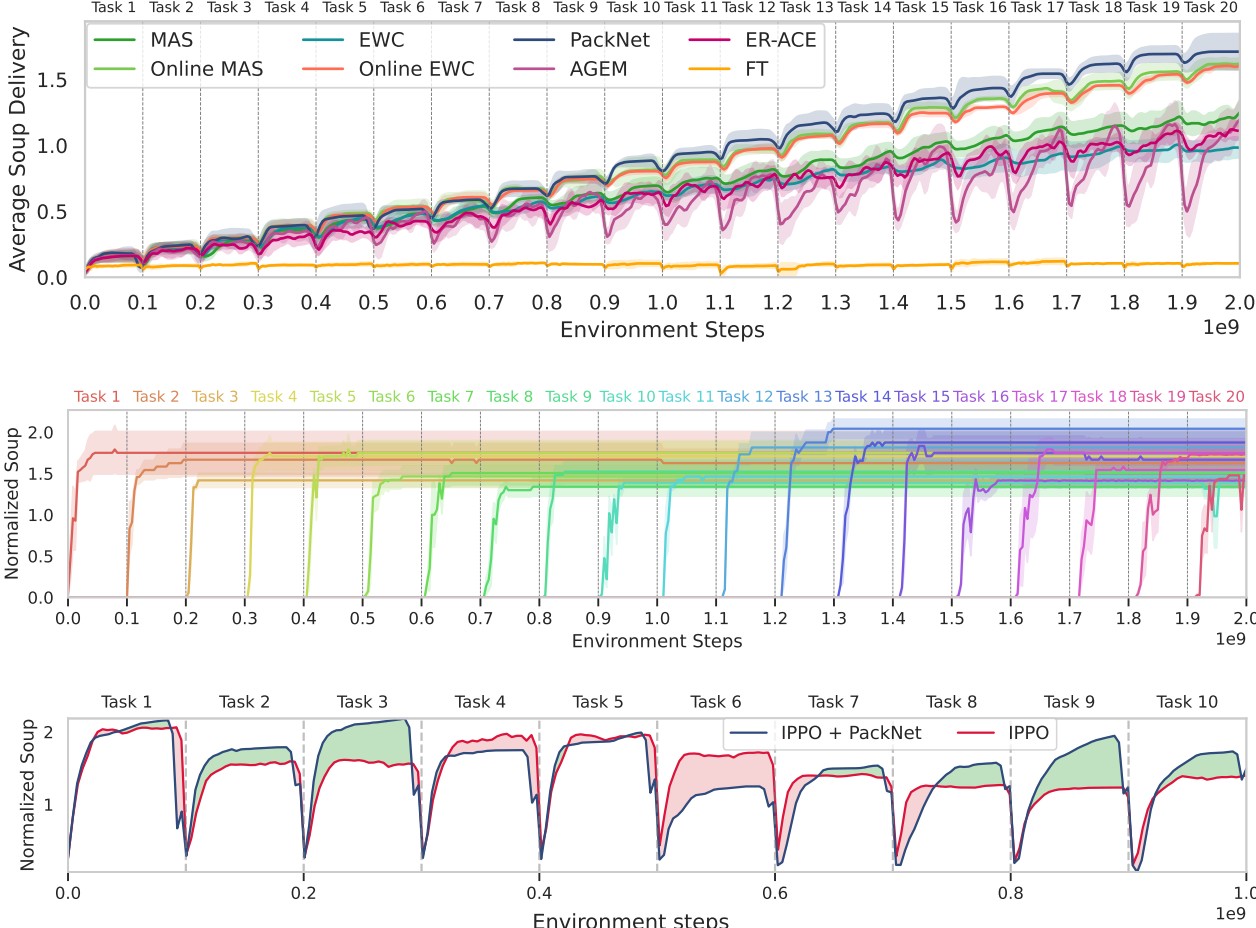

*Figure 3.* **Top:** The soup delivery evaluation curves on Level 1 display the performance gap across baselines. **Middle**: Online EWC displays almost no forgetting on Level 1. After learning a task, performance stays stable until the very end. **Bottom**: By leveraging knowledge from previously learned tasks, PackNet achieves higher soup deliveries during training than IPPO trained from scratch on each task. Green shading between the curves indicates positive forward transfer, red indicates negative.

later ones. For each task $i < N$, let $\tau_i = i \cdot \Delta$ denote the timestep at which training on task $i$ ends, and let $s_i^\star = s_i(\tau_i)$ be the score achieved at that point. For $t > \tau_i$, we define a normalized performance drop $d_i(t)$ and apply an exponentially decaying weight $w_i(t)$, which penalizes earlier forgetting more strongly:

$$d_i(t) = \max\left(0, \frac{s_i^\star - s_i(t)}{s_i^\star}\right), \quad w_i(t) = e^{-\lambda \frac{t-\tau_i}{T-\tau_i}}. \quad (2)$$

We aggregate these weighted drops over time and average across tasks to obtain the overall forgetting score:

$$\mathcal{F} = \frac{1}{N-1} \sum_{i=1}^{N-1} \frac{\sum_{t>\tau_i} w_i(t)\, d_i(t)}{\sum_{t>\tau_i} w_i(t)}. \quad (3)$$

**Forward Transfer** Forward transfer measures how prior experience affects the speed at which new tasks are learned. We compare training performance against a single-task baseline. For each task $i$, we compute the normalized area under

the learning curve (AUC) during its training window for both the continual learner and the baseline:

$$\text{AUC}_i = \frac{1}{\Delta} \int_{(i-1)\Delta}^{i\Delta} s_i(t)\, dt, \quad \text{AUC}_i^{\text{b}} = \frac{1}{\Delta} \int_0^{\Delta} s_i^{\text{b}}(t)\, dt. \quad (4)$$

The forward transfer score is then defined as the normalized difference between these AUCs, where positive values indicate accelerated learning due to prior experience:

$$\mathcal{FT} = \frac{1}{N} \sum_{i=1}^{N} \frac{\text{AUC}_i - \text{AUC}_i^{\text{b}}}{1 - \text{AUC}_i^{\text{b}}}. \quad (5)$$

These metrics capture performance well, but they say little about the *division of labor* between agents. Two agents can deliver the same number of soups while swapping roles, or with one agent doing all the work. The score stays flat while the coordination underneath it shifts. In Appendix L, we devise complementary settings and metrics to study coordination and role specialization.

*Table 2.* Comparison of CL methods in combination with IPPO across three difficulty levels with 95% confidence intervals. PackNet attains the highest average score $\mathcal{S}$ at every level while keeping forgetting $\mathcal{F}$ near zero, whereas AGEM achieves the best forward transfer $\mathcal{FT}$ at the cost of high forgetting. Best value per column is depicted in **bold**.

| Method | Level 1 | | | Level 2 | | | Level 3 | | |
|---|---|---|---|---|---|---|---|---|---|
| | $\mathcal{S}\uparrow$ | $\mathcal{F}\downarrow$ | $\mathcal{FT}\uparrow$ | $\mathcal{S}\uparrow$ | $\mathcal{F}\downarrow$ | $\mathcal{FT}\uparrow$ | $\mathcal{S}\uparrow$ | $\mathcal{F}\downarrow$ | $\mathcal{FT}\uparrow$ |
| FT | $0.05_{\pm0.00}$ | $0.70_{\pm0.10}$ | $-0.32_{\pm0.02}$ | $0.05_{\pm0.02}$ | $0.72_{\pm0.24}$ | $-0.36_{\pm0.02}$ | $0.01_{\pm0.02}$ | $0.59_{\pm0.12}$ | $-0.59_{\pm0.09}$ |
| EWC | $1.00_{\pm0.10}$ | $0.02_{\pm0.04}$ | $-0.58_{\pm0.03}$ | $0.98_{\pm0.11}$ | $0.03_{\pm0.03}$ | $-0.61_{\pm0.04}$ | $0.62_{\pm0.17}$ | $0.01_{\pm0.01}$ | $-0.71_{\pm0.05}$ |
| MAS | $1.28_{\pm0.13}$ | $0.03_{\pm0.03}$ | $-0.49_{\pm0.06}$ | $1.07_{\pm0.01}$ | $0.01_{\pm0.01}$ | $-0.55_{\pm0.04}$ | $0.81_{\pm0.04}$ | $0.00_{\pm0.00}$ | $-0.67_{\pm0.02}$ |
| Online EWC | $1.61_{\pm0.03}$ | $0.00_{\pm0.00}$ | $-0.21_{\pm0.02}$ | $1.34_{\pm0.03}$ | $0.01_{\pm0.01}$ | $-0.30_{\pm0.03}$ | $1.13_{\pm0.11}$ | $0.02_{\pm0.02}$ | $-0.39_{\pm0.09}$ |
| Online MAS | $1.63_{\pm0.07}$ | $0.03_{\pm0.03}$ | $-0.15_{\pm0.03}$ | $0.76_{\pm0.09}$ | $0.22_{\pm0.06}$ | $-0.08_{\pm0.02}$ | $0.53_{\pm0.05}$ | $0.29_{\pm0.01}$ | $-0.16_{\pm0.01}$ |
| AGEM | $1.17_{\pm0.21}$ | $0.41_{\pm0.04}$ | $\mathbf{0.04}_{\pm0.01}$ | $1.00_{\pm0.15}$ | $0.33_{\pm0.04}$ | $\mathbf{0.00}_{\pm0.02}$ | $0.76_{\pm0.34}$ | $0.41_{\pm0.06}$ | $\mathbf{-0.08}_{\pm0.05}$ |
| ER-ACE | $1.10_{\pm0.07}$ | $0.19_{\pm0.02}$ | $-0.24_{\pm0.02}$ | $0.88_{\pm0.05}$ | $0.25_{\pm0.03}$ | $-0.29_{\pm0.05}$ | $0.65_{\pm0.04}$ | $0.37_{\pm0.00}$ | $-0.43_{\pm0.02}$ |
| PackNet | $\mathbf{1.75}_{\pm0.13}$ | $\mathbf{0.00}_{\pm0.00}$ | $-0.05_{\pm0.05}$ | $\mathbf{1.38}_{\pm0.11}$ | $\mathbf{0.00}_{\pm0.00}$ | $-0.21_{\pm0.03}$ | $\mathbf{1.29}_{\pm0.04}$ | $\mathbf{0.00}_{\pm0.00}$ | $-0.33_{\pm0.01}$ |

# 5. Experiments

We train each task $\mathcal{T}_i$ for $\Delta = 10^8$ environment steps with dense rewards. Following (Wołczyk et al., 2021) and (Tomilin et al., 2023), we keep the task identity known during training and evaluation. We evaluate the policy after every 100 updates by running 10 evaluation episodes on all tasks in the sequence. Throughout the paper, error bars, shaded regions, and table $\pm$ ranges denote 95% confidence intervals computed across the five seeds. Our experiments are conducted on a dedicated compute node with a 72-core 3.2 GHz AMD EPYC 7F72 CPU and a single NVIDIA H100 GPU. The wall-clock time to train a task for $10^8$ steps on our hardware is $\sim$ 2 minutes. We adopt many of JaxMARL's default settings for our network configuration, IPPO setup, and training processes. For exact hyperparameters and setup details, see Appendix C.2.

## 5.1. Baseline Comparison

We evaluate MEAL against a set of widely used CL baselines. Fine-Tuning (**FT**) is a naive baseline where the policy is trained sequentially across tasks without any mechanism to prevent forgetting. **EWC** (Kirkpatrick et al., 2017) penalizes changes to important parameters, with importance measured using the Fisher Information Matrix. **Online EWC** extends this by maintaining a running estimate of parameter importance. **MAS** (Aljundi et al., 2018) computes importance based on how parameters influence the policy's output, rather than gradients. We analogously introduce **Online MAS**, maintaining a running estimate in the same spirit as Online EWC. **AGEM** (Chaudhry et al., 2018) is a replay-based method that projects the current gradient update to avoid interference with past tasks, using a memory buffer of stored experiences. **ER-ACE** (Caccia et al., 2022) applies an asymmetric loss to current and buffered data, balancing plasticity and stability within a replay-based framework. **PackNet** (Mallya & Lazebnik, 2018) incrementally allocates portions of the network to each task through pruning and freezing.

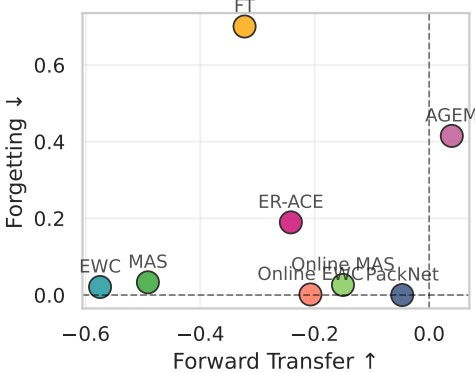

*Figure 4.* Forward transfer and forgetting on Level 1 reveal a clear **stability–plasticity trade-off**. Regularization-based methods (EWC, MAS, and their online variants) minimize forgetting but at the cost of negative forward transfer. AGEM and FT sit at the opposite extreme, exhibiting high forgetting, but remaining plastic. PackNet strikes the best balance, combining low forgetting with near-zero forward transfer.

As the core MARL algorithm, we use **IPPO** (De Witt et al., 2020), which integrates seamlessly with all model-free CL methods and has been shown to perform strongly on SMAC and Overcooked. We additionally evaluate **MAPPO** (Yu et al., 2022), a centralized-training variant of PPO that conditions each agent's critic on the global state and **HAPPO** (Kuba et al., 2022), which extends MAPPO with sequential per-agent updates to preserve a monotonic improvement guarantee.

Figure 3 (top) compares our baselines on Level 1, and Table 2 reports the exact metrics for all levels. Fine-Tuning (FT) has no CL mechanism and immediately forgets a task once it is left behind. AGEM achieves the best forward transfer, the only positive or near-zero values in the table, but pays for it with high forgetting. Regularization-based methods sit at the opposite end, retaining past tasks well while showing strongly negative forward transfer. PackNet offers the best overall balance. It attains the highest average score at every difficulty level and keeps forgetting

*Table 3.* IPPO + Online EWC with varying numbers of agents on Levels 1 and 2. Performance peaks at two agents on Level 1 and three on Level 2, as agents split the workload and act in parallel, then declines as the team grows. Larger teams are harder to learn and retain, with forgetting rising steeply at four or more agents on Level 2 as agents interfere with one another.

| Agents | Level 1 | | | Level 2 | | |
|---|---|---|---|---|---|---|
| | $\mathcal{S}\uparrow$ | $\mathcal{F}\downarrow$ | $\mathcal{FT}\uparrow$ | $\mathcal{S}\uparrow$ | $\mathcal{F}\downarrow$ | $\mathcal{FT}\uparrow$ |
| 1 Agent | 1.03 | 0.18 | **-0.08** | 0.97 | 0.13 | **-0.14** |
| 2 Agents | **1.61** | **0.03** | -0.21 | 1.34 | **0.06** | -0.30 |
| 3 Agents | 1.49 | 0.21 | -0.35 | **1.43** | 0.10 | -0.35 |
| 4 Agents | 1.27 | 0.19 | -0.46 | 0.86 | 0.82 | -0.27 |
| 5 Agents | 0.96 | 0.27 | -0.48 | 1.03 | 0.69 | -0.33 |

near zero, with forward transfer closer to zero than the regularization methods. ER-ACE falls between these groups, forgetting less than FT and AGEM, while retaining more plasticity than the regularization-based methods. Among the regularizers, the online variants clearly outperform their cumulative counterparts on average score, most notably Online EWC, which remains strong even on Level 3. Figure 3 (middle) visualizes Online EWC's per-task stability. A deeper analysis is provided in Appendix I.1. Per-task evaluation curves for all baselines are shown in Figure 25. As difficulty increases, performance degrades across all methods, with the cumulative regularizers (EWC, MAS) declining most sharply (Figure 10). Figure 4 further illustrates this stability–plasticity trade-off underlying this decline.

## 5.2. $N$-Agent MEALs

MEAL supports arbitrary team sizes, letting us measure how the number of agents affects CMARL. We run IPPO[2] + Online EWC with 1-5 agents. A single agent delivers fewer soups because it cannot parallelize the workflow (Table 3). With a second agent, one can refill the pot with onions while the other plates and delivers, so throughput rises. The gain does not continue indefinitely. Performance peaks at two agents on Level 1 and three on Level 2, then falls as the team keeps growing. The later peak on Level 2 reflects its larger layouts, which leave room for an extra agent to work in parallel before the kitchen becomes crowded.

Beyond the peak, additional agents hurt rather than help. Since IPPO trains independent policies in what remains a joint MDP, each added agent enlarges the joint action space, worsens non-stationarity as teammates update at once, and muddies credit assignment under the shared reward. These pressures are sharpest in the continual setting, as retention and transfer become harder. Without explicit communication or role allocation, IPPO cannot learn and retain coordination as the team grows. We conclude that CL becomes harder with every added agent.

---

[2]With one agent, IPPO reduces to standard PPO.

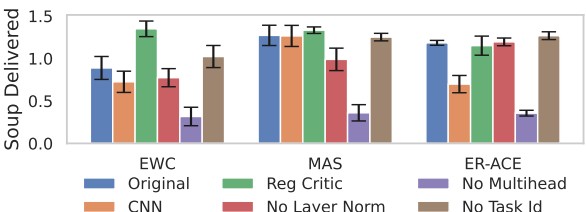

*Figure 5.* **Ablation study**. Removing multi-head outputs causes the largest performance drop, while the task ID has a negligible effect. MAS performs equally well with the CNN encoder, while EWC and ER-ACE prefer an MLP. Critic regularization positively affects EWC. Layer normalization helps MAS retain performance.

## 5.3. Ablation Study

To determine which components are crucial for CMARL on MEAL, we ablate five components on EWC, MAS, and ER-ACE: multi-head architectures, task identity inputs, critic regularization, layer normalization, and replacing the MLP with a CNN encoder. The results in Figure 5 show that multi-head outputs are by far the most critical component. Removing them collapses performance to roughly a third of the baseline for all methods, likely due to uncontrolled interference between tasks in the shared output head. In contrast, the one-hot encoded task ID vector has a negligible effect, changing performance only within the confidence intervals. Prior CRL studies (Wołczyk et al., 2021; Tomilin et al., 2023) report that it is beneficial to regularize only the actor and let the critic adapt freely. In our setting, however, regularizing the critic as well improves performance, most clearly for EWC. Removing layer normalization slightly hurts all methods, indicating that stabilizing activations across tasks mitigates harmful scale drift under continual regularization. Finally, swapping to a CNN encoder hurts EWC and ER-ACE while leaving MAS largely unchanged. Given the small layouts (6×6 to 7×7), CNNs struggle to extract meaningful features and add unnecessary parameter overhead, making simple MLPs a better fit in this setting.

## 5.4. Impact of Task Sequence Length

Short task sequences can give a misleading picture of method performance. To illustrate this, we compare standard EWC and Online EWC on Level 1 sequences of length $|\mathcal{T}| \in \{10, 100\}$. As seen in Table 5, across 10 tasks, both methods appear similarly stable, since the accumulated Fisher penalty in EWC has not yet become overly restrictive. Over 100 tasks, however, standard EWC increasingly over-regularizes the shared backbone, limiting plasticity and leading to earlier performance saturation, whereas Online EWC retains capacity by exponentially decaying past importance weights. This example shows how certain dynamics in CRL only emerge over long horizons, where saturation and drift become clear. This raises the question of how many method rankings in prior work would change under longer sequences. See Appendix I for an extended discussion.

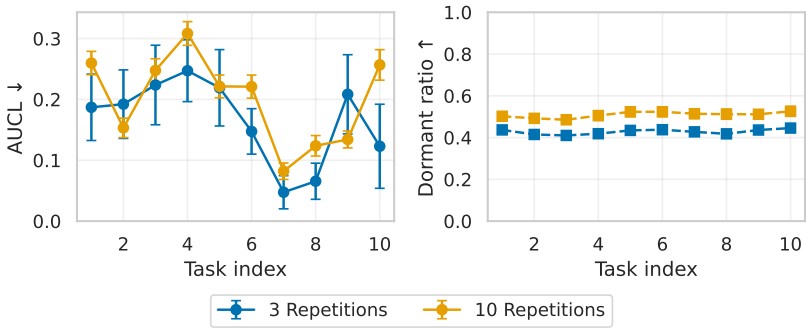

*Table 4.* Averaged plasticity metrics on Level 1. The AUC-loss increases more sharply between 1 and 3 repetitions than between 3 and 10, indicating that learning capacity degrades more early on. Dormancy, however, increases more gradually across repetitions, with no early spike.

| Reps | AUCL $\downarrow$ | Dormancy $\downarrow$ |
|---|---|---|
| 1 | $0.000 \pm 0.000$ | $0.408 \pm 0.003$ |
| 3 | $0.166 \pm 0.052$ | $0.428 \pm 0.006$ |
| 10 | $0.201 \pm 0.018$ | $0.509 \pm 0.022$ |

*Figure 6.* **Loss of plasticity in MEAL.** AUCL (**left**) captures performance loss, and the Dormancy ratio (**right**) quantifies the fraction of inactive neurons. Increasing the repetition count leads to lower performance and more dormant neurons.

*Table 5.* EWC and Online EWC on 10-task and 100-task Level 1 sequences. While both methods perform similarly on short sequences, their behavior diverges heavily on longer sequences.

| Method | 10 tasks $\mathcal{S} \uparrow$ | 100 tasks $\mathcal{S} \uparrow$ |
|---|---|---|
| IPPO + EWC | $1.60_{\pm 0.08}$ | $0.23_{\pm 0.03}$ |
| IPPO + Online EWC | $1.55_{\pm 0.02}$ | $1.45_{\pm 0.05}$ |

### 5.5. Loss of Network Plasticity

A well-documented pitfall in continual RL is the gradual loss of **plasticity**, an agent's ability to fit new data after many tasks (Abbas et al., 2023; Dohare et al., 2024). To test whether MEAL exhibits the same pathology, we train IPPO on a Level 1 10-task sequence over multiple repetitions and compare the performance between them. We track two metrics: (i) **AUC-loss** captures capacity drop, (ii) **Dormancy ratio** quantifies the fraction of inactive neurons in the policy network. Appendix G describes these metrics in greater detail and depicts the training curves. We observe that both metrics deteriorate with longer training (Table 4 and Figure 6), confirming that loss of plasticity also appears in the multi-agent setting. Despite our setting spanning over 10B environment steps, well beyond the scale of prior studies (Abbas et al., 2023; Dohare et al., 2024), MEAL exhibits a notably smaller loss of plasticity. We conjecture that this is due to the use of multiple output heads that isolate task-specific outputs, mitigate gradient interference, and enable the shared backbone to learn more transferable features.

### 5.6. Partially Observable MEALs

Although Overcooked is fully observable by design, we introduce a partially observable variant to better reflect real-world sensing constraints (limited field of view, occlusions). Following popular MARL environments (Resnick et al., 2018; Mohanty et al., 2020; Agapiou et al., 2022; Ellis et al., 2023), each agent receives an egocentric, direction-aware observation window with all outside tiles masked. The spec-

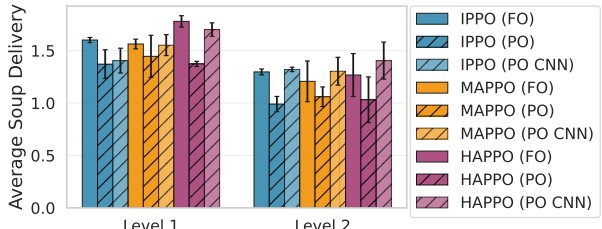

*Figure 7.* Across IPPO, MAPPO, and HAPPO, partial observability (PO) lowers average soup delivery relative to full observability (FO), but switching the MLP encoder to a CNN (PO CNN) recovers most of the loss, exceeding the FO baseline on Level 2. Encoder choice interacts strongly with observability.

ification and difficulty scaling of this window are detailed in Appendix D. We run 20-task sequences with Online EWC under partial observability (PO) and compare against the fully observable (FO) setting, across IPPO, MAPPO, and HAPPO.

A consistent pattern emerges across all three algorithms (Figure 7). Moving from full to partial observability lowers the soup delivery, as masking distant tiles complicates credit assignment and destabilizes value targets. Crucially, this loss is largely recoverable: replacing the MLP encoder with a CNN restores performance close to, and on Level 2 often above, the fully observable baseline. The effect is strongest on Level 2, where the larger layouts make local spatial structure more informative, and the CNN's spatial inductive bias lets agents extract it from the egocentric window. This contrasts with the fully observable setting (Section 5.3), where CNNs hurt because the small fully visible grids offered little spatial structure to exploit. Under partial observability, that inductive bias becomes an asset rather than overhead.

### 5.7. Continual Partner Adaptation

MEAL also supports continual adaptation to changing co-operation partners rather than changing layouts. We train an ego agent against a fixed sequence of eight diverse partners, allotting $10^8$ steps per partner. Since each partner is frozen,

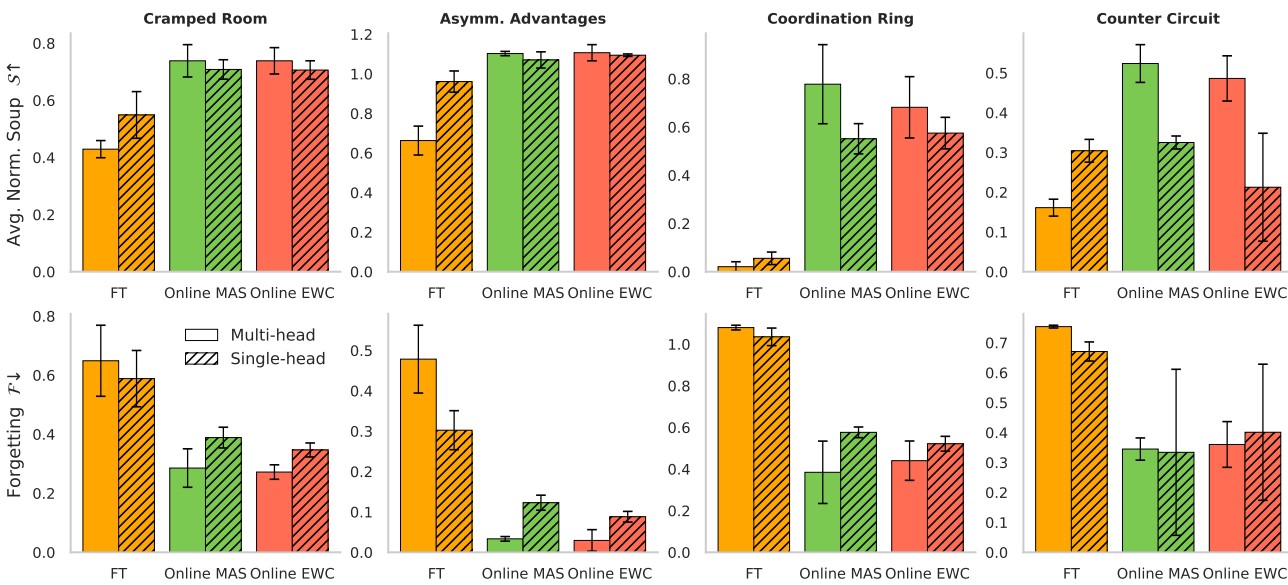

*Figure 8.* **Continual partner adaptation** across four original Overcooked layouts. An ego agent sequentially adapts to eight diverse partners, $10^8$ steps each. **Top**: average normalized soup delivery ($\mathcal{S} \uparrow$). **Bottom**: forgetting ($\mathcal{F} \downarrow$). Solid bars denote multi-head policies, hatched bars single-head. Multi-head outputs help the regularization-based methods, but not FT.

only the ego agent learns, so we use PPO in place of a MARL algorithm. We additionally compare multi-head and single-head policies to test whether the architectural finding from Section 5.3 carries over to partner adaptation. All other settings follow the main experimental setup. Appendix M describes the partners and their ordering in full.

Figure 8 reports the average score and forgetting on four Overcooked layouts. FT forgets sharply whenever a partner is replaced and barely delivers soup on *Coordination Ring*. Online EWC and Online MAS retain far more and reach higher scores, with the largest gap on the harder Coordination Ring layout. Unlike the layout experiments, where multi-head outputs were the single most important component (Section 5.3), here their benefit depends on the CL method. They improve both score and retention for Online EWC and Online MAS, most clearly on the harder layouts, but give no benefit to FT, where the single-head policy matches or beats it. Multi-head outputs appear to pay off only alongside a mechanism that protects the shared backbone. EWC and MAS regularize the trunk, so separate heads let each partner specialize without interference, whereas FT rewrites the trunk regardless, and the extra heads only dilute its updates.

## 6. Conclusion

Continual RL experiments have been computationally demanding due to their inherent sequential training regime and the need to repeatedly evaluate performance across previously seen tasks. As a result, most prior work has been restricted to short task sequences. Crucially, we showed that experimental conclusions can change substantially as the number of tasks grows, suggesting the need for efficient benchmarks. MEAL addresses this through a hardware-accelerated, fully JAX-based pipeline, with on-demand procedural generation that scales task difficulty and supports long sequences. Our study shows that combining CL methods with MARL algorithms works well in simple settings with multi-head architectures, but degrades as layouts grow, sequences lengthen, agents are added, rewards get sparser, and partial observability is imposed. Most notably, increasing the number of agents makes tasks harder both to learn and to retain, establishing continual cooperation as a challenge distinct from single-agent retention, and one that adapting to novel partners makes harder still. We see immediate headroom for methods that (i) are purpose-built for CMARL, jointly handling partner- and environment-level non-stationarity, (ii) stabilize credit assignment under partial observability across long sequences, and (iii) promote structured exploration and robust coordination. We believe MEAL offers a practical foundation for studying these challenges at scale.

## Acknowledgments

This work was conducted with the assistance of the Dutch national e-infrastructure, generously supported by the SURF Cooperative under grant EINF-12816. The authors also thank the International Max Planck Research School for Intelligent Systems (IMPRS-IS) for supporting C. Ruhdorfer.

## Impact Statement

This paper introduces a benchmark for continual multi-agent reinforcement learning. Its main societal benefit is reducing the computational cost of this research: by enabling long task sequences to run in hours on a single GPU rather than requiring large clusters, MEAL lowers the energy footprint of experiments and broadens access for researchers without substantial compute. The experiments use a simple cooking game and carry no direct risks. More broadly, methods for agents that adapt over time and coordinate with others could eventually inform real-world multi-agent systems, where reliable cooperation and graceful adaptation matter; we view MEAL as a step toward studying these properties carefully before such systems are deployed. We do not foresee specific harms arising from this work.

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

# A. Implementation Details

This section provides additional technical details for the MEAL benchmark implementation. We first describe the heuristic used to estimate an upper bound on the number of soups that can be delivered in a layout. We then outline the procedural kitchen generator used to create diverse environments and the validator that ensures generated layouts remain solvable.

## A.1. Maximum Soup Delivery Calculator

Let a kitchen layout $\mathcal{L}$ be defined by four disjoint sets of tiles (onion piles $\mathcal{O}$, plate piles $\mathcal{P}$, pots $\mathcal{K}$, delivery counters $\mathcal{G}$) and a set of walls $\mathcal{W}$. A tile $(x, y)$ is *walkable* if $(x, y) \notin \mathcal{W}$.

**Neighbourhood of an object family.** We denote the set of walkable tiles adjacent (in the 4-neighbour sense) to any object in $\mathcal{S}$ as:

$$\mathcal{N}(\mathcal{S}) = \{(x', y') \mid (x, y) \in \mathcal{S}, \ \|(x', y') - (x, y)\|_1 = 1, \ (x', y') \notin \mathcal{W}\}$$

**Shortest obstacle-aware distance.** Given two tile sets $A, B \subseteq \mathbb{Z}^2$, we define

$$d(A, B) = \min_{a \in A, \, b \in B} \ \text{dist}^{\mathcal{G}_{\mathcal{L}}}_{\text{manhattan}}(a, b),$$

where $\mathcal{G}_{\mathcal{L}}$ is the grid graph induced by walkable tiles. We realize this via a breadth-first search (BFS).

**Single-agent cook–deliver cycle.** A soup requires three onions, one plate pick-up, one soup pick-up, and one delivery. Let

$$d_{\text{onion}} = d\big(\mathcal{N}(\mathcal{O}), \mathcal{N}(\mathcal{K})\big), \quad d_{\text{plate}} = d\big(\mathcal{N}(\mathcal{P}), \mathcal{N}(\mathcal{K})\big), \quad d_{\text{goal}} = d\big(\mathcal{N}(\mathcal{K}), \mathcal{N}(\mathcal{G})\big).$$

The optimistic *movement cost* for one cycle is

$$c_{\text{move}} = 3\, d_{\text{onion}} + d_{\text{plate}} + 1 + d_{\text{goal}} + 3.$$

**Interaction overhead.** Every pick-up or drop is assumed to take a constant $c_{\text{act}} = 2$ steps (turn + interact). With $n_{\text{int}} = 3 \times 2 + 1 + 1 + 1 = 9$ interactions per cycle, the overhead is $c_{\text{over}} = n_{\text{int}}\, c_{\text{act}} = 18$.

**Cycle time and upper bound.** Including the fixed cooking time $c_{\text{cook}} = 20$ steps, the single-agent cycle time is

$$T_{\text{cycle}} = c_{\text{move}} + c_{\text{cook}} + c_{\text{over}}.$$

For an episode horizon $H$, we upper-bound the number of soups by

$$N_{\max}(\mathcal{L}, H) = \lfloor H/T_{\text{cycle}} \rfloor,$$

and convert it to reward with $r_{\text{deliver}} = 20$:

$$R_{\max}(\mathcal{L}, H) = 20\, N_{\max}(\mathcal{L}, H).$$

The bound assumes *a single agent acting optimally*. It ignores multi–agent collaboration and therefore *underestimates* throughput in layouts where multiple agents can parallelize the workflow. Listing 1 contains the exact implementation.

## A.2. Procedural Kitchen Generator

**Objective.** Given a random seed and user-selectable parameters (number of agents $n_a$, layout height range $[h_{\min}, h_{\max}]$, layout width range $[w_{\min}, w_{\max}]$, and wall-density $\rho$), the goal is to emit a *solvable* grid string $G$ representing the `Overcooked` environment.

### A.2.1. NOTATION

Let $h, w \sim \text{UniformInt}(h_{\min}, h_{\max})$, $\text{UniformInt}(w_{\min}, w_{\max})$, and denote by $\mathcal{C} = \{(i, j) \mid 1 \leq i \leq h - 2, \ 1 \leq j \leq w - 2\}$ the set of *internal* cells (outer walls excluded). Its cardinality is $N_{\text{int}} = (h - 2)(w - 2)$. An *unpassable* cell contains either a hard wall (#) or an interactive tile; we write $N_{\text{unpass}}(G)$ for the number of such cells in $G$.

**Listing 1** Heuristic upper bound (`calculate_max_soup`).

```python
# overcooked_upper_bound.py      (excerpt)
COOK_TIME = 20
ACTION_OVERHEAD = 2
INTERACTIONS_PER_CYCLE = 3 * 2 + 1 + 1 + 1
OVERHEAD_PER_CYCLE = INTERACTIONS_PER_CYCLE * ACTION_OVERHEAD

def calculate_cycle_time(layout, n_agents=2):
    ...
    move_cost = 3 * d_onion + d_plate + 1 + d_goal + 3
    return move_cost + COOK_TIME + OVERHEAD_PER_CYCLE

def calculate_max_soup(layout, episode_len, n_agents=2):
    cyc = calculate_cycle_time(layout, n_agents)
    soups = episode_len // cyc
    return int(soups)
```

### A.2.2. ALGORITHM

The generator performs the following loop until a valid grid is produced (Listing 2):

1. **Draw size.** Sample $h, w$ and create an $h \times w$ matrix initialised to FLOOR tiles, then overwrite the border with WALL.

2. **Place interactive tiles.** For each symbol in {GOAL, POT, ONION_PILE, PLATE_PILE} choose a random multiplicity $m \in \{1, 2\}$ and stamp the symbol onto $m$ uniformly chosen floor cells.

3. **Inject extra walls.** Let $n_{\text{target}} = \lceil \rho\, N_{\text{int}} \rceil$ and $n_{\text{add}} = \max\big(0,\, n_{\text{target}} - N_{\text{unpass}}(G)\big)$. Place $n_{\text{add}}$ additional walls on random floor cells.

4. **Place agents.** Stamp $n_a$ AGENT symbols on random remaining floor cells.

5. **Validate.** Run the deterministic `evaluate_grid` solver; if it returns `True`, terminate and return $(G)$, otherwise restart.

6. **Cleanup.** Remove any interactive elements and tiles that are unreachable from all agent positions.

7. **Return.** Output the final grid.

**Solvability criterion.** The validator (Appendix A.3) checks (i) path connectivity between every agent and each interactive tile family, (ii) at least one pot reachable from an onion pile and a plate pile, and (iii) at least one goal reachable from a pot. This is implemented via multiple breadth-first searches. Appendix A.3 further details the evaluator logic.

**Wall-density effect.** Because interactive tiles themselves count as obstacles, the algorithm first places them, then *only as many extra walls as needed* to reach the prescribed obstacle ratio $\rho$. This keeps difficulty roughly constant even when two copies of every station are spawned.

**Failure handling.** If any placement stage exhausts the pool of empty cells, or the validator rejects the grid, the attempt is aborted and restarted with a fresh $h, w$ sample. We cap retries at `max_attempts` (default 2000); empirically fewer than five attempts suffice for $\rho \leq 0.3$.

**Complexity.** All placement operations are $O(hw)$ in the worst case (linear scans to collect empty cells), while validation runs a constant number of BFS passes, each $O(hw)$. Hence one successful attempt is $O(hw)$.

**Listing 2** Overcooked Layout Generator

```python
def generate_random_layout(seed, params):
    rng = random.Random(seed)
    for attempt in range(params.max_attempts):
        h = rng.randint(*params.h_range)
        w = rng.randint(*params.w_range)
        grid = init_floor_with_border(h, w)

        # 1. Interactive tiles
        for sym in [GOAL, POT, ONION_PILE, PLATE_PILE]:
            if not place_random(grid, sym, rng.randint(1, 2), rng):
                break  # restart

        # 2. Extra walls to hit density
        n_target = round(params.wall_density * (h-2)*(w-2))
        n_add = n_target - count_unpassable(grid)
        if not place_random(grid, WALL, n_add, rng):
            continue  # restart

        # 3. Agents
        if not place_random(grid, AGENT, params.n_agents, rng):
            continue

        # 4. Validate
        if evaluate_grid(to_string(grid)):
            return to_string(grid)
```

## A.3. Layout Validator

We guarantee that every procedurally generated kitchen is *playable* by running a deterministic validator before training begins. The validator implements ten checks, ranging from basic grid sanity to cooperative reachability. A grid is accepted only if **all** checks pass.

**Notation.** Let $G$ be an $h \times w$ character matrix with symbols $\{\texttt{W}, \texttt{X}, \texttt{O}, \texttt{B}, \texttt{P}, \texttt{A}, \textvisiblespace\}$ for walls, delivery, onion pile, plate pile, pot, agent, and floor. Interactive tiles are $\mathcal{I} = \{\texttt{X}, \texttt{O}, \texttt{B}, \texttt{P}\}$, and unpassable tiles $\mathcal{U} = \mathcal{I} \cup \{\texttt{W}\}$.

**Validation rules.**

**R1** *Rectangularity* – all rows have equal length.

**R2** *Required symbols* – each of W,X,O,B,P,A appears at least once.

**R3** *Border integrity* – every outer-row/column tile is in $\{\texttt{W}\} \cup \mathcal{I}$.

**R4** *Interactivity access* – every tile in $\mathcal{I} \cup \{\texttt{A}\}$ has at least one 4-neighbour that is A or floor.

**R5** *Reachable onions* – at least one onion pile is reachable by some agent.

**R6** *Usable pots* – at least one pot is reachable *and* lies in the same connected component as a reachable onion.

**R7** *Usable delivery* – at least one delivery tile is reachable *and* lies in a component with a usable pot.

**R8** *Agent usefulness* – each agent can either interact with an object directly or participate in a hand-off (adjacent wall shared with the other agent's region).

**R9** *Coverage* – the union of agents' reachable regions touches every object family in $\mathcal{I}$.

**R10** *Handoff counter* – if one agent cannot reach all families, a wall tile adjacent to *both* regions exists, enabling item transfer.

Rules R5–R10 rely on two depth-first searches (DFS) from the agent positions. The DFS explores floor and agent tiles only; whenever it touches an interactive tile, that family is marked as "found." Let $\text{Reach}_k \subseteq [h] \times [w]$ denote tiles reached from agent $k$ ($k \in \{1, 2\}$).

**Algorithmic outline.** Listing 3 shows a condensed version of the validator.

---

**Listing 3** Condensed Layout Validator.

---

```python
def validate(grid_str):
    g = [list(r) for r in grid_str.splitlines()]
    h, w = len(g), len(g[0])

    # R1{R3 omitted for brevity ...

    # Depth-first search from a start cell
    def dfs(i, j, seen):
        if (i, j) in seen or g[i][j] in UNPASSABLE_TILES - {AGENT}:
            return
        seen.add((i, j))
        for di, dj in ((1,0),(-1,0),(0,1),(0,-1)):
            dfs(i+di, j+dj, seen)

    # Agents and family reachability
    a1, a2 = [(i, j) for i,r in enumerate(g)
                     for j,c in enumerate(r) if c == AGENT]
    reach1, reach2 = set(), set()
    dfs(*a1, reach1); dfs(*a2, reach2)

    # Helper: reachable(\mathcal{S}, reach)
    def any_reach(symbols, reach):
        return any(g[i][j] in symbols for i,j in reach)

    # R5{R7
    if not any_reach({ONION_PILE}, reach1|reach2):        return False
    if not any_reach({POT}, reach1|reach2):               return False
    if not any_reach({GOAL}, reach1|reach2):              return False

    # R8{R10 (usefulness & hand-off)
    def useful(reach_me, reach_other):
        # direct or shared-wall hand-off
        for i,j in reach_me:
            if g[i][j] in INTERACTIVE_TILES: return True
            if g[i][j] == FLOOR and any(
                (abs(i-i2)+abs(j-j2) == 1 and g[i2][j2] == WALL)
                for i2,j2 in reach_other):
                return True
        return False

    if not useful(reach1, reach2): return False
    if not useful(reach2, reach1): return False
    return True
```

---

**Complexity.** All checks are $O(hw)$ and require only two DFS traversals, thus one validation runs in time linear to the grid area and is negligible compared with policy learning.

**Practical impact.** In practice, fewer than 1% of generator attempts fail validation when wall-density $\rho \leq 0.15$ and kitchen size $\geq 8 \times 8$. We therefore cap retries at 2000 without noticeable overhead.

# B. Environment Specifications

This section describes the environment interface used in MEAL. We outline the *Overcooked* environment dynamics, the observation representation provided to agents, the discrete action space, and the dense/sparse reward functions.

**Dynamics**    Agents act synchronously at each time step. Moves into walls or occupied tiles are no-ops, and simultaneous swaps are disallowed (both agents remain in place). Agents can `interact` with the tile they are facing, which deterministically updates the object's state (pick/place, add onion, plate, deliver). The pots initiate a fixed cook timer of $c_{cook}=20$ steps when the third onion is added, and the soup can only be plated after it has finished cooking.

**Observations**    Each agent receives a fully observable grid-based observation of shape $(H, W, 26)$, where $H$ and $W$ are the height and width of the grid, and the 26 channels encode tile types (e.g., walls, agents, onions, plates, pots, delivery stations) and states (e.g., cooking progress, held item). To maintain a consistent observation space for CL, we fix the shape to $(H_{max}, W_{max}, 26)$, where $H_{max}$ and $W_{max}$ are the largest grid dimensions in the sequence, and pad all smaller layouts with walls.

**Action Space**    At each timestep, all agents select one of six discrete actions from a shared action space $\mathcal{S} = \{$up, down, left, right, stay, interact$\}$. Movement actions translate the agent forward if the target tile is free (i.e., not a wall or occupied), while `stay` maintains the current position. The `interact` action is context-dependent and allows agents to pick up or place items, add ingredients to pots, serve completed dishes, or deliver them at the goal location. Importantly, there is no built-in communication action; all coordination emerges from environment interactions.

**Rewards**    Agents receive a shared team reward: $r_t = r_{deliver} + r_{onion} \cdot \mathbb{1}_{onion\_in\_pot} + r_{plate} \cdot \mathbb{1}_{plate\_pickup} + r_{soup} \cdot \mathbb{1}_{soup\_pickup}$, where $r_{deliver} = 20$ is the reward for delivering soup, and the other terms provide shaped rewards for intermediate progress. We include two reward settings: in the **sparse** setting, $r_{onion} = r_{plate} = r_{soup} = 0$; in the **dense** setting, $r_{onion} = r_{plate} = 3$, and $r_{soup} = 5$. We compare these two reward settings in Appendix N.

# C. Experimental Setup

This section describes the experimental configuration used in the MEAL benchmark. We first outline the neural network architectures used for the actor–critic policy. We then report the hyperparameters that remain fixed across experiments, including PPO training settings and continual learning–specific parameters.

## C.1. Network Architecture

All agents share the same actor–critic backbone, implemented in `Flax`. Two encoder variants are provided:

- **MLP** (default) : observation tensor is flattened to a vector and passed through 2 fully-connected layers of width 128.
- **CNN** : three 32-channel convolutions with kernel sizes 5×5, 3×3, 3×3 feed a 64-unit projection, followed by a single 128-unit dense layer.

Common design knobs (controlled from the CLI) are:

- **Activation** (`relu` *vs.* `tanh`).
- **LayerNorm** : applied after every hidden layer when `use_layer_norm` is enabled.
- **Shared vs. Separate encoder** : with `shared_backbone` the two heads operate on a common representation; otherwise actor and critic keep independent trunks.
- **Multi-head outputs** : if `use_multihead` is set, each head holds a distinct slice of logits/values for every task ($\text{num\_tasks} = |\mathcal{T}|$). The correct slice is selected with a cheap tensor reshape.
- **Task-one-hot conditioning** : setting `use_task_id` concatenates a one-hot vector of length $|\mathcal{T}|$ before the actor/critic heads, mimicking "oracle" task identifiers used in many CL papers.

All linear/conv layers use orthogonal weight initialisation with gain $\sqrt{2}$ (or 0.01 for policy logits) and zero biases. The policy outputs a `distrax.Categorical`; the critic outputs a scalar.

*Table 6.* Fixed hyper-parameters. All experiments use dense reward shaping, two agents, and IPPO unless noted. CL coefficients $\lambda$ refer to the regularization strength passed to each method.

| Parameter | Value |
|---|---|
| *Optimization (PPO)* | |
| Activation | ReLU |
| Optimizer | Adam (Optax) |
| Adam $(\beta_1, \beta_2)$ | $(0.9,\ 0.999)$ |
| Adam $\epsilon$ | $10^{-5}$ |
| Weight decay | none |
| Learning rate $\eta$ | $10^{-3}$ |
| LR annealing | linear $(10^{-3} \rightarrow 10^{-4})$ |
| Env. steps per task $\Delta$ | $10^8$ |
| Parallel envs | 2048 |
| Rollout length $T$ | 400 |
| Effective batch size | $2048 \times 400 = 819\,200$ |
| Updates per task | $\lfloor 10^8/819\,200 \rfloor = 122$ |
| Update epochs | 8 |
| Minibatches per update | 16 |
| Gradient steps per task | $122 \times 8 \times 16 = 15{,}616$ |
| Discount $\gamma$ | 0.99 |
| GAE $\lambda$ | 0.95 |
| PPO clip $\epsilon$ | 0.2 |
| Entropy coef. $\alpha_{\text{ent}}$ | 0.01 |
| Value-loss coef. $\alpha_{\text{vf}}$ | 0.5 |
| Max grad-norm | 1.0 |
| *Continual-learning specifics* | |
| Sequence length $|\mathcal{T}|$ | 20 (base sequence), repeated $r$ times |
| Reg. coefficient $\lambda$ | $10^{11}$ (EWC), $10^9$ (MAS), $10^7$ (L2) |
| Online EWC/MAS decay | 0.9 |
| Importance episodes / steps | 5 / 500 |
| Regularize critic / heads | No / No |
| AGEM Memory size | 100 000 transitions |
| AGEM Sample size (per proj.) | 1024 |
| *Miscellaneous* | |
| Reward shaping horizon | $2.5 \times 10^6$ steps (linear to 0) |
| Evaluation interval | every 5 policy updates |
| Evaluation episodes | 10 |
| Random seeds | $\{1\,..\,5\}$ |

## C.2. Hyperparameters

Table 6 lists settings that are *constant* across every experiment unless stated otherwise. At task boundaries, the optimizer state and policy parameters are carried over, the rollout buffers are reset, and the JAX RNG is advanced.

*Table 7.* Field-of-view specification for the partially observable MEAL variant. Window size and directional extents scale with difficulty.

| Difficulty | Grid Size | Forward View | Side View | Rear View | Obs Window (H×W) |
|---|---|---|---|---|---|
| Easy | 6–7 | 1 | 1 | 0 | 2×3 |
| Medium | 8–9 | 2 | 1 | 0 | 3×3 |
| Hard | 10–11 | 3 | 2 | 1 | 3×5 |

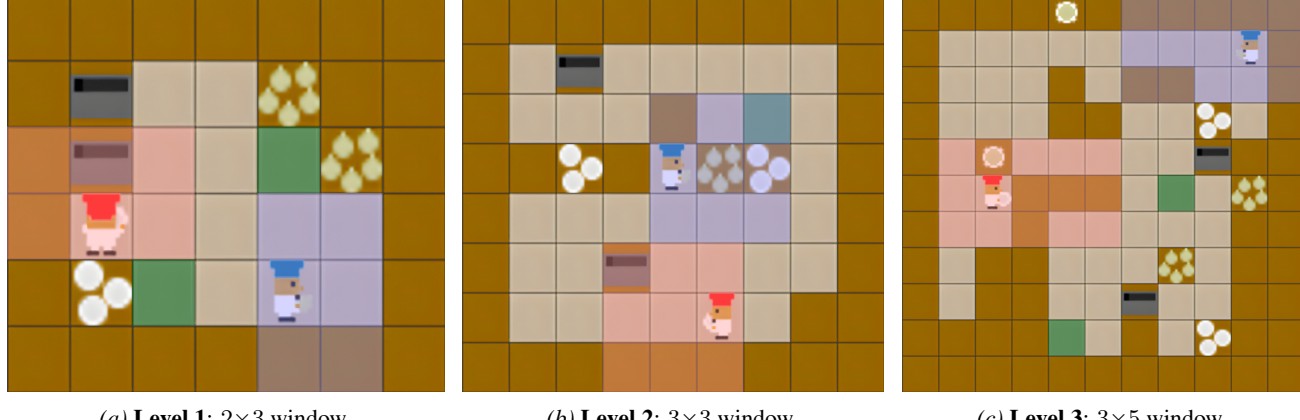

*(a)* **Level 1**: 2×3 window.      *(b)* **Level 2**: 3×3 window.      *(c)* **Level 3**: 3×5 window.

*Figure 9.* Egocentric observation windows by difficulty. Visibility grows with difficulty but remains partial, preserving the need for exploration and memory.

## D. Partially Observable MEALs

To more closely mimic the constraints faced by real-world agents, we introduce a *direction-aware* egocentric observation setting. Each agent perceives a rectangular window centered on itself, with tiles outside this window masked. The window is anisotropic with respect to the agent's heading: we separate forward, side, and rear extents, which increase with difficulty (Table 7). This scaling is intentionally balanced with the overall environment design: as the grid size grows with difficulty, the perceptual window also expands to maintain a comparable challenge-to-information ratio. Consequently, the tasks become POMDPs, where exploration, memory (e.g., recurrent state), and implicit/explicit coordination provide tangible benefits. In particular, Level 1 removes rear context entirely, Level 2 extends the look-ahead by one tile, and Level 3 adds both longer look-ahead and rear visibility, reducing blind spots while preserving partial observability (Figure 9).

## E. Difficulty Levels

Higher levels of difficulty in MEAL pose greater challenges for both learning and retention. As the grid size and obstacle density increase, the environment becomes more complex: interactable items are farther apart, and navigation paths are longer and more convoluted. This increases the number of steps required to complete a recipe. Not only does this make learning each task harder, but it also forces the agent to retain and execute longer action sequences to successfully complete a recipe. Higher-level layouts also add demands for plasticity and transfer. The larger layout space introduces greater variability between tasks, making it harder to reuse learned behavior. These factors collectively lead to lower performance as difficulty increases, as shown in Figure 10. Online EWC has a steady upward trend on Level 1, while on higher levels, the performance gap becomes more evident as the number of tasks increases.

## F. Curriculum Learning

In all training settings, agents consistently struggle on Level 3 tasks with large grids. Curriculum learning has been shown to improve final performance on difficult tasks by gradually increasing task complexity (Bengio et al., 2009; Narvekar et al., 2020; Portelas et al., 2020). We investigate whether a simple difficulty-based curriculum can help agents better learn harder MEAL tasks under the same data budget. To this end, we design a curriculum sequence where each difficulty level contributes an equal number of tasks. Specifically, we sample 5 layouts each from Level 1 (easy), Level 2 (medium), and

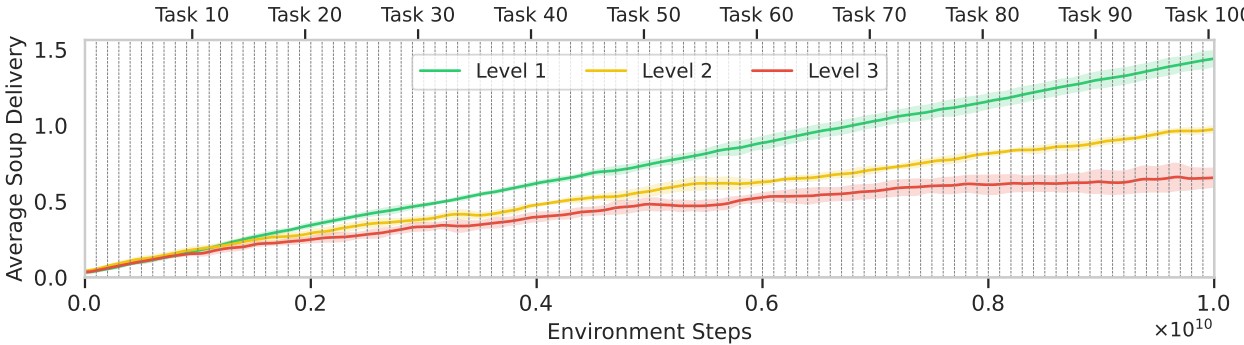

*Figure 10.* **Average Soup Delivery** over the course of training Online EWC on a sequence of 100 generated tasks per difficulty level. Shaded regions indicate 95% confidence intervals across 5 seeds. The clear gap between levels shows the effectiveness of MEAL's simple difficulty level design.

Level 3 (hard), and present them in ascending order of difficulty (easy 1-5, medium 6-10, hard 11-15). We compare this curriculum setting against the default training setup, where agents are trained on sequences containing tasks from a single difficulty level only. We report results only on the corresponding task window. For example, medium-level performance is measured on tasks 6–10 in the curriculum sequence and compared against tasks 6-10 from a 10-task sequence consisting exclusively of Level 2 tasks.

The results in Table 8 show no statistically significant difference between the two strategies on Level 2, given the high variance. However, on Level 3, the curriculum strategy nearly doubles performance. A plausible explanation is that, under curriculum training, the agent first experiences 5 easy and 5 medium tasks, where it receives denser reward signals and more frequent successes. This exposure likely builds useful priors and stabilizes learning, improving adaptation to harder tasks later. In contrast, the default strategy trains only on hard tasks throughout the sequence, where exploration is more challenging and initial rewards are more difficult to obtain, leading to weaker performance overall.

*Table 8.* Curriculum vs. default training under an equal data budget. We report the Average Soup Delivery over the task windows of the respective difficulty.

| Strategy | Medium (6–10) | Hard (11–15) |
|----------|---------------|--------------|
| Default | $0.693 \pm 0.147$ | $0.328 \pm 0.238$ |
| Curriculum | $0.668 \pm 0.152$ | $\mathbf{0.653 \pm 0.181}$ |

## G. Network Plasticity

Loss of neural network plasticity is a well-studied phenomenon in continual RL, where agents gradually become less able to adapt to new tasks as training progresses. A number of metrics have been proposed to characterize this effect, typically measuring how updates propagate through the network or how parameter sensitivity changes over time.

### G.1. Metrics

We follow Abbas et al. (2023); Dohare et al. (2024) and quantify **plasticity**, the ability to fit fresh data after many tasks, by three complementary metrics computed from the training reward.

**Notation.** For a single task let $r_t$ be the online reward at step $t \le T$. A repetition experiment presents the same task $R$ times, so the trace splits into $R$ contiguous segments of equal length $L = T/R$. We smooth $r_t$ with a Gaussian kernel (bandwidth $\sigma$) and define the cumulative average

$$\bar{r}(t) = \frac{1}{t} \sum_{i=1}^{t} r_i, \qquad t = 1, \dots, L.$$

All metrics compare a later repetition $j > 0$ with the *baseline* repetition $j = 0$.

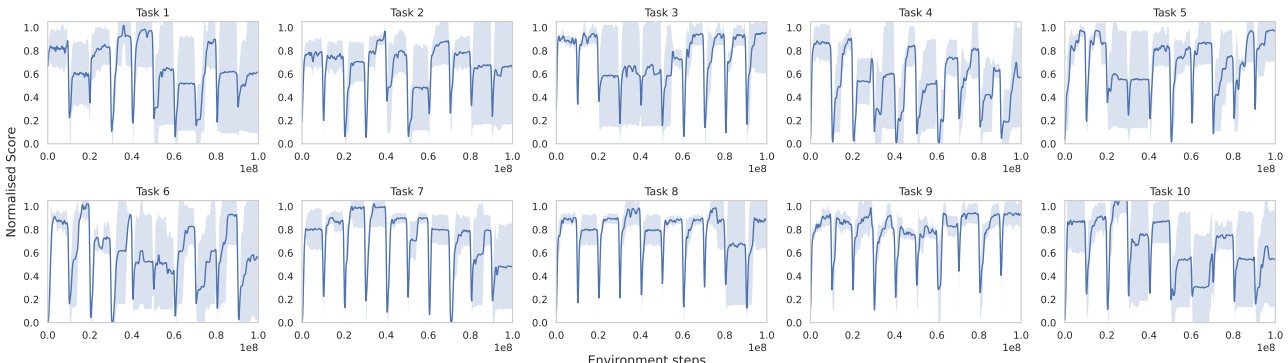

*Figure 11.* Training curves of FT across a Level 1 10-task sequence repeated ten times over 5 seeds.

*Table 9.* All sequence-averaged metrics for FT with 95% confidence intervals.

| Repeats | AUC-loss ↓ | Dormant Ratio ↓ | FPR ↑ | RAUC ↑ |
|---|---|---|---|---|
| 1 | $0.000 \pm 0.000$ | $0.408 \pm 0.003$ | $1.000 \pm 0.000$ | $1.000 \pm 0.000$ |
| 3 | $0.166 \pm 0.052$ | $0.428 \pm 0.006$ | $0.926 \pm 0.111$ | $0.901 \pm 0.114$ |
| 10 | $0.201 \pm 0.018$ | $0.509 \pm 0.022$ | $0.891 \pm 0.066$ | $0.872 \pm 0.070$ |

**AUC–loss.** Let $\mathrm{AUC}_j = \int_0^L \bar{r}_j(t)\, dt$. The capacity drop for repetition $j$ is

$$\mathrm{loss}_j \;=\; 1 - \frac{\mathrm{AUC}_j}{\mathrm{AUC}_0}, \qquad j = 1, \dots, R-1, \tag{6}$$

where 0 indicates perfect retention. We report the mean of Eq. (6) over repetitions and seeds.

**Dormant Neuron Ratio.** Following Sokar et al. (2023), we also measure *dormancy*, the fraction of units that remain effectively inactive during training. Given hidden activations $h \in \mathbb{R}^{B \times H}$ for batch size $B$ and layer width $H$, we compute the mean absolute activation per unit $m = \frac{1}{B}\sum_{b=1}^{B} |h_{b,:}|$. Normalizing by the global mean $\bar{m} = \frac{1}{H}\sum_{j=1}^{H} m_j$, we obtain scores $s_j = m_j/(\bar{m} + \epsilon)$. A unit is considered *dormant* if $s_j \leq \tau$ for some threshold $\tau$ (we use $\tau = 0.01$). The Dormant Neuron Ratio is the fraction of dormant units, averaged across layers and seeds. Higher values indicate more inactive capacity, and hence reduced plasticity.

**Final-Performance Ratio (FPR).** With $p_j = \bar{r}_j(L-1)$ the plateau reward of repetition $j$,

$$\mathrm{FPR}_j \;=\; \frac{p_j}{p_0}, \qquad j = 1, \dots, R-1, \tag{7}$$

so $\mathrm{FPR}_j > 1$ implies no loss, $\mathrm{FPR}_j < 1$ indicates degraded plateau performance.

**Raw-AUC Ratio (RAUC).** Using the *unsmoothed* running reward,

$$\mathrm{RAUC}_j \;=\; \frac{\mathrm{AUC}_j^{\mathrm{raw}}}{\mathrm{AUC}_0^{\mathrm{raw}}}, \qquad j = 1, \dots, R-1, \tag{8}$$

which captures the total reward accumulated during learning. Higher values in Eq. (7)–Eq. (8) are better.

**Sequence-level aggregation.** For a task sequence of length $|\mathcal{T}|$ we compute the per-task means of (6)–(8) and average across tasks, yielding a single global score per repetition count $R$.

### G.2. Training Curves

Figure 11 plots the mean normalized score of the fine-tuning (FT) baseline over ten repetitions. Performance on Tasks 8 and 9 remains virtually unchanged, indicating little to no plasticity loss. In contrast, Tasks 1, 2, 6, and 10 show a clear

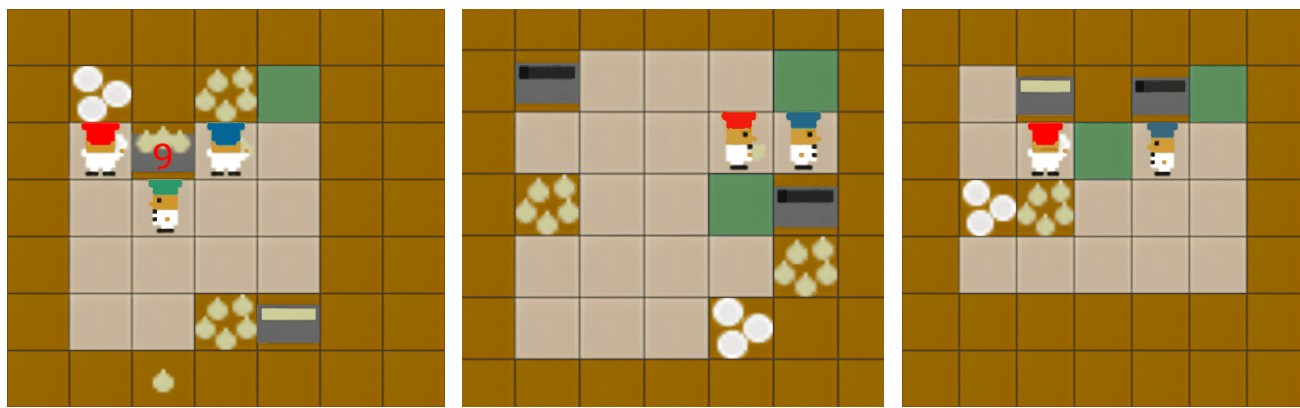

*(a)* **Single-pot fixation**. All agents are clustered around a single pot, waiting for it to finish cooking, while ignoring the fully-cooked soup in the bottom pot, ready to be plated.

*(b)* **Deadlock**. The red agent wishes to place an onion in the right pot, but is blocked by the blue agent standing in the corner, who is unable to step aside.

*(c)* **Role collapse.** The red agent completes the pipeline solo while the blue agent wanders without contributing. The learned policy has settled on a local minimum.

*Figure 12.* Qualitative failure modes observed in Overcooked. All behaviors stem from inadequate coordination, limited exploration, or poor role allocation.

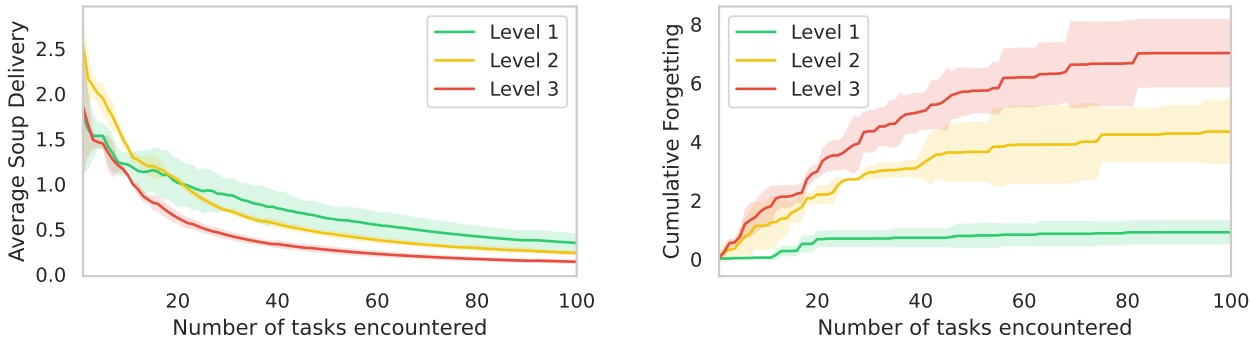

*Figure 13.* EWC over 100 tasks. **Left**: gradual decline in average score as more tasks are encountered. **Right**: on higher levels, forgetting increases more rapidly.

degradation: the agent fails to recover the score achieved during the first repetition, illustrating a pronounced loss of plasticity.

## H. Common Pitfalls

Despite dense rewards and simple layouts, we frequently observed learned policies falling into a few recurring failure modes that throttle throughput and coordination. Figure 12 illustrates three we noticed repeatedly across layouts, levels, and methods, though we did not quantify their frequency. All three stem from a failure to maintain stable coordination. Agents fixate on a single pot, deadlock in shared chokepoints, or collapse into one agent doing the work while the other idles.

## I. Long Task Sequences

Prior work in continual reinforcement learning has largely focused on evaluating only a small number of tasks sequentially. In this section, we therefore explore how increasing the number of tasks in a sequence affects CL performance. To the best of our knowledge, evaluating on a 100-task continual RL sequence has not previously been reported in the literature. Figure 13 shows that performance gradually declines as the number of tasks grows. The performance gap across difficulty levels is especially pronounced for forgetting. Training for 10 billion environment steps across 100 tasks took $\sim$ 4 hours on a single GPU.

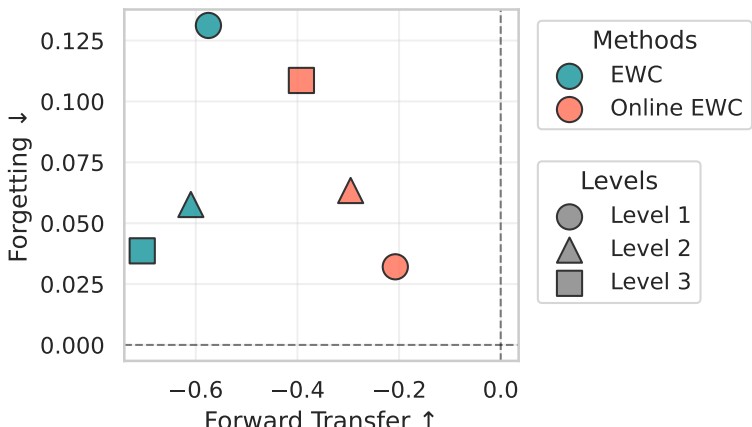

*Figure 14.* Comparison of EWC and Online EWC across all difficulty levels on 20-task sequences, evaluated in terms of forward transfer and forgetting. Each marker denotes a method's performance at a given level. Online EWC consistently achieves higher forward transfer than EWC across all levels. Forgetting for Online EWC grows with difficulty, while for EWC it decreases.

### I.1. EWC vs. Online EWC: A Case Study

EWC accumulates importance over *all* past tasks and penalizes drift along high-Fisher directions with a fixed quadratic. Online EWC maintains a *running*, exponentially decayed Fisher, emphasizing recent tasks and relaxing old constraints. Both use the same heads, meaning that the penalty acts on the shared trunk. When layouts are small, not only are the tasks easier to learn, but the same features are more likely to work across tasks. Strong anchoring preserves those features, curbing forgetting and yielding a higher average score. The stability–plasticity trade-off is favorable because plasticity demands are modest. This trade-off is visualized in Figure 14, where Online EWC shows higher forward transfer at every level, indicating it adapts to new layouts more readily. Notably, Online EWC's forgetting rises with difficulty, while EWC's decreases. The cumulative Fisher penalty of Online EWC pays off on small Level 1–2 layouts, but underfits on Level 3 since harder layouts demand larger representation shifts. By contrast, Online EWC uses a decayed Fisher that down-weights older tasks and manages to keep enough plasticity to learn the new layouts. Level 3 forces longer paths, bottlenecks, and role specialization, which require larger representational updates. EWC's cumulative constraints over-tighten the trunk and slow adaptation, while Online EWC's decay frees capacity for those shifts, so it learns the hard tasks more effectively. The multiple output heads alone are not enough. They isolate outputs, but the penalty sits on the shared backbone. When the trunk needs to be rewired for new Level 3 tasks, EWC resists too much, while Online EWC allows it more. Moreover, credit assignment is noisier on Level 3 due to sparser effective signals and longer horizons. A single, stale Fisher snapshot can misdirect EWC's penalty. The rolling estimate in Online EWC smooths that noise and tracks the current regime more closely.

Scaling up the number of tasks to 100 amplifies the behavioral difference between these methods. As shown in Figure 15, standard EWC quickly saturates: its performance plateaus around 20 tasks, while Online EWC continues to improve throughout the sequence. The difference arises from how the two methods accumulate and apply their regularization terms. EWC optimizes the loss

$$\mathcal{L}_{\text{EWC}} = \mathcal{L}_{\text{task}} + \frac{\lambda}{2} \sum_{t=1}^{k-1} \sum_{i} F_{t,i} \left( \theta_i - \theta_{t,i}^* \right)^2 , \tag{9}$$

where each Fisher matrix $F_t$ captures parameter importance after task $t$. The regularizer grows with every task, anchoring the network more tightly to older solutions. Consequently, plasticity decays over time, and adaptation to new tasks becomes progressively harder. Online EWC instead compresses past information through an exponentially decayed Fisher:

$$F_{\text{online}}^{(k)} = \gamma F_{\text{online}}^{(k-1)} + F_k, \tag{10}$$

where the decay factor $\gamma \in (0, 1)$ controls how quickly the influence of older tasks fades to restore the capacity for new

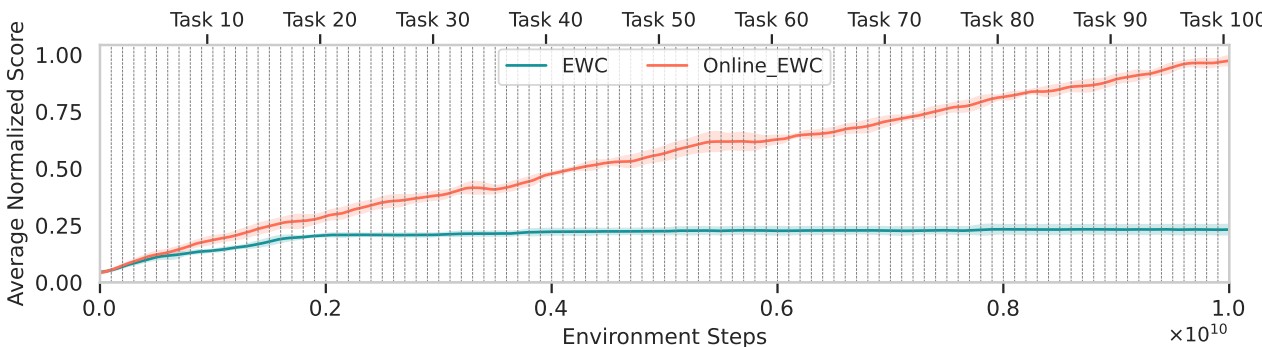

*Figure 15.* Average normalized performance of EWC and Online EWC over a 100-task sequence (Level 1). The standard EWC variant continuously accumulates regularization terms from all previous tasks, leading to excessive constraint and early performance saturation around 20 tasks. In contrast, Online EWC compresses past information with a decay factor, maintaining plasticity and achieving sustained learning throughout the sequence.

ones. This running Fisher approximation is then used to define a single consolidated quadratic penalty:

$$\mathcal{L}_{\text{Online EWC}} = \mathcal{L}_{\text{task}} + \frac{\lambda}{2} \sum_i F_{\text{online},i}^{(k)} \left( \theta_i - \theta_{k,i}^* \right)^2 . \tag{11}$$

Over long sequences, these mechanisms diverge sharply: standard EWC eventually over-regularizes, effectively freezing the shared backbone. The model retains early knowledge but cannot repurpose features as the environment shifts. Online EWC's decayed Fisher avoids this over-constraining and sustains meaningful adaptation even after dozens of tasks. The continued improvement across 100 tasks illustrates that certain dynamics in continual reinforcement learning only emerge over *long horizons*, where saturation and drift become clear.

This case study reinforces one of the core motivations behind **MEAL**: short sequences often fail to reveal such discrepancies. We therefore urge the continual RL community to focus on longer streams of tasks, where stability–plasticity trade-offs are more likely to truly emerge.

## J. Algorithm Comparison

The main paper pairs every CL method with IPPO. To test whether the choice of underlying MARL algorithm changes which CL methods work, we re-run all eight CL methods on Level 1 with two alternative algorithms: MAPPO, which conditions each agent's critic on the global state, and HAPPO, which extends MAPPO with sequential per-agent updates. Figure 16 reports the resulting average performance for every CL method and MARL algorithm pairing.

Two patterns stand out. First, the relative ordering of CL methods is largely preserved across algorithms: FT fails everywhere, Online EWC, Online MAS, and PackNet are consistently strong, and AGEM and ER-ACE sit in the middle. The choice of MARL algorithm does not overturn the conclusions drawn from IPPO in the main paper. Second, the algorithms are not interchangeable. HAPPO yields the highest scores for most of the regularization-based methods, lifting EWC from 1.00 under IPPO to 1.47 and Online EWC from 1.61 to 1.80, the best single result in the grid. PackNet, by contrast, peaks under IPPO (1.75) and gains nothing from the more elaborate algorithms, and AGEM degrades sharply away from IPPO, dropping from 1.17 to around 0.67 under both MAPPO and HAPPO.

A plausible reading is that methods which protect the shared backbone benefit from HAPPO's more stable, coordinated updates, whereas PackNet, which already isolates per-task capacity through pruning, has little to gain and is most effective with the simplest algorithm. Overall, the comparison suggests that while IPPO is a reasonable default, the best CL method and MARL algorithm should be chosen together rather than in isolation.

| | FT | EWC | MAS | Online EWC | Online MAS | AGEM | ER-ACE | PackNet |
|---|---|---|---|---|---|---|---|---|
| IPPO | 0.05 ±0.00 | 1.00 ±0.10 | 1.28 ±0.13 | 1.61 ±0.03 | 1.63 ±0.07 | 1.17 ±0.21 | 1.10 ±0.07 | 1.75 ±0.13 |
| MAPPO | 0.09 ±0.02 | 1.08 ±0.20 | 0.82 ±0.28 | 1.57 ±0.05 | 1.58 ±0.05 | 0.67 ±0.18 | 1.09 ±0.18 | 1.61 ±0.06 |
| HAPPO | 0.10 ±0.00 | 1.47 ±0.18 | 1.61 ±0.05 | 1.80 ±0.05 | 1.74 ±0.09 | 0.68 ±0.08 | 1.19 ±0.06 | 1.63 ±0.14 |

*Figure 16.* **Algorithm comparison** on Level 1. Average Soup Delivery $\mathcal{S}$ (with 95% confidence intervals) for each CL method (columns) paired with each MARL algorithm (rows: IPPO, MAPPO, HAPPO). Darker cells indicate higher performance. HAPPO is the strongest algorithm for most regularization-based methods, while PackNet peaks under IPPO.

*Table 10.* Online EWC under additional sources of non-stationarity on 20-task sequences, reported as average soup delivery $\mathcal{S}$ with 95% confidence intervals. Each factor is applied in isolation; **Combined** applies all four simultaneously.

| Dynamics | Level 1 | Level 2 | Level 3 |
|---|---|---|---|
| Default | $1.61_{\pm 0.03}$ | $1.34_{\pm 0.03}$ | $1.13_{\pm 0.11}$ |
| Pot Size | $1.59_{\pm 0.08}$ | $1.28_{\pm 0.04}$ | $1.02_{\pm 0.05}$ |
| Soup Timer | $1.55_{\pm 0.11}$ | $1.27_{\pm 0.03}$ | $1.06_{\pm 0.05}$ |
| Sticky Actions | $1.48_{\pm 0.03}$ | $1.18_{\pm 0.03}$ | $0.84_{\pm 0.11}$ |
| Slippery Tiles | $1.37_{\pm 0.03}$ | $1.16_{\pm 0.02}$ | $0.83_{\pm 0.05}$ |
| **Combined** | $1.12_{\pm 0.07}$ | $0.95_{\pm 0.03}$ | $0.67_{\pm 0.04}$ |

## K. Non-Stationary MEALs

MEAL's primary task sequences vary the kitchen layout or partners, which already constitute valid continual RL settings. Layout and partner changes are not the only axis of non-stationarity the environment can express, however, and incorporating others enriches the benchmark with new evaluation settings and further opportunities to study how CL methods behave. We introduce four additional factors, each randomized or activated per task in the sequence.

**Pot Size.** By default, each soup requires three onions. We instead randomize this requirement between one and five onions for each pot at the start of every task.

**Soup Timer.** By default, a pot takes 20 game ticks to cook. We randomize this duration between 10 and 30 ticks for each pot at the start of every task.

**Sticky Actions.** Inspired by the Arcade Learning Environment (Machado et al., 2018), each agent repeats its previous action with some probability rather than executing the one it selected. This probability scales with difficulty: 10% on Level 1, 20% on Level 2, and 30% on Level 3.

**Slippery Tiles.** In each task, 25% of the walkable tiles are designated emphslippery. When an agent steps on a slippery tile, it may slip on the following step and move to a random adjacent tile instead. The slip probability scales with difficulty: 35% on Level 1, 50% on Level 2, and 65% on Level 3.

We probe each factor in isolation and then test all four combined, training Online EWC on 20-task sequences across all three difficulty levels (Table 10). Individually, each source of non-stationarity mildly reduces performance, with the perturbations to agent movement (sticky actions and slippery tiles) hurting more than those to the recipe (pot size and soup timer).

Combined, their effects accumulate and substantially increase task difficulty, lowering the average score by roughly a third at every level.

## L. Coordination Analysis

A central claim of this work is that retaining cooperation is a challenge distinct from retaining task performance. This section gathers the experiments that examine cooperation directly, rather than through aggregate soup delivery. We approach it from two directions. First, we add environment settings that make coordination unavoidable, either by partitioning agents so no one can cook alone (Appendix L.1) or by assigning complementary roles that split the recipe between them (Appendix L.2). These stress whether CL methods can learn and retain genuinely cooperative strategies under controlled conditions. Second, we introduce metrics that measure how agents coordinate, independent of how much soup they deliver (Appendix L.3), revealing that cooperative structure can erode even while the score holds steady.

### L.1. Forced Coordination

In standard Overcooked, whether agents end up in separate regions of the kitchen is left to chance: depending on the generated layout, two agents may share a fully connected space and complete soups independently, or they may be split across counters and forced to cooperate. This means the degree of required coordination varies uncontrollably from task to task. To study coordination under reliably demanding conditions, we add an explicit setting that guarantees agents are partitioned and must rely on one another.

Concretely, we add a `forced_coordination` flag to the layout generator. When enabled, the validator additionally ensures that (1) no walkable path exists between any pair of agents, and (2) for every agent, at least one of the three cook–deliver cycle components (onion↔pot, pot↔plate, pot↔delivery) is unreachable within that agent's region, while the union of all agents' reachable regions still admits a valid cycle. The maximum-soup estimator (Appendix A.1) is adapted accordingly, finding the best cook–deliver cycle within the union of all agents' reachable regions. As a result, no single agent can complete a soup alone, and progress depends on handing off ingredients and dishes across counters.

We evaluate Online EWC on 20-task sequences with two agents (Table 11). On Level 1, forced coordination barely affects performance, as the compact layouts make hand-offs easy to discover. The gap widens with difficulty, reaching a substantial drop on Level 3. Inspecting the training curves and gameplay recordings, we find that the larger, more partitioned layouts produce more tasks where the agents fail to establish any working hand-off, leaving some soups undelivered.

*Table 11.* Online EWC with and without forced coordination on 20-task sequences with two agents, reported as average soup delivery $\mathcal{S}$ with 95% confidence intervals. Forced coordination makes a valid soup impossible for any single agent, requiring cross-counter hand-offs.

| Forced Coordination | Level 1 | Level 2 | Level 3 |
|---|---|---|---|
| Off | $1.61_{\pm 0.03}$ | $1.34_{\pm 0.03}$ | $1.13_{\pm 0.11}$ |
| On | $1.59_{\pm 0.13}$ | $1.17_{\pm 0.05}$ | $0.82_{\pm 0.21}$ |

### L.2. Designated Roles

In Overcooked, agents are identical in their capabilities and attributes. However, in many real-world scenarios, autonomous agents either 1) possess different physical properties or 2) are functionally identical but are expected to fulfill distinct, complementary roles to cooperate effectively for a common goal. To capture this dimension in MEAL, we design a heterogeneous agent setting with **designated roles**.

In this variant, two agents are randomly assigned one of the two predefined roles at the start of each task: **chef** and **waiter**. The chef is responsible for preparing the soup by loading onions into the pot, but cannot pick up plates. The waiter handles dish delivery but cannot pick up onions. This enforces complementary capabilities, meaning neither agent can complete the full recipe alone, meaning that successful catering requires coordinated role execution and adaptation. Note that the roles are sampled per task and may switch across tasks, making continual learning essential.

We evaluate this setting over 20-task Level 1 sequences using EWC with IPPO under shared rewards. Table 12 compares the heterogeneous setup to the default homogeneous setting. We observe a clear performance drop in the role-restricted setting, as throughput decreases when agents are limited to certain actions and cannot flexibly switch between tasks. Another factor

*Table 12.* Homogeneous vs. heterogeneous (designated roles) 2-agent training results over Level 1 sequences using shared rewards and IPPO + EWC.

| Agents | $\mathcal{S}\uparrow$ | $\mathcal{F}\downarrow$ | $\mathcal{FT}\uparrow$ |
|---|---|---|---|
| Homogeneous | **0.90 ± 0.04** | **0.01 ± 0.01** | **0.20 ± 0.08** |
| Heterogeneous | 0.68 ± 0.09 | 0.03 ± 0.02 | −0.05 ± 0.09 |

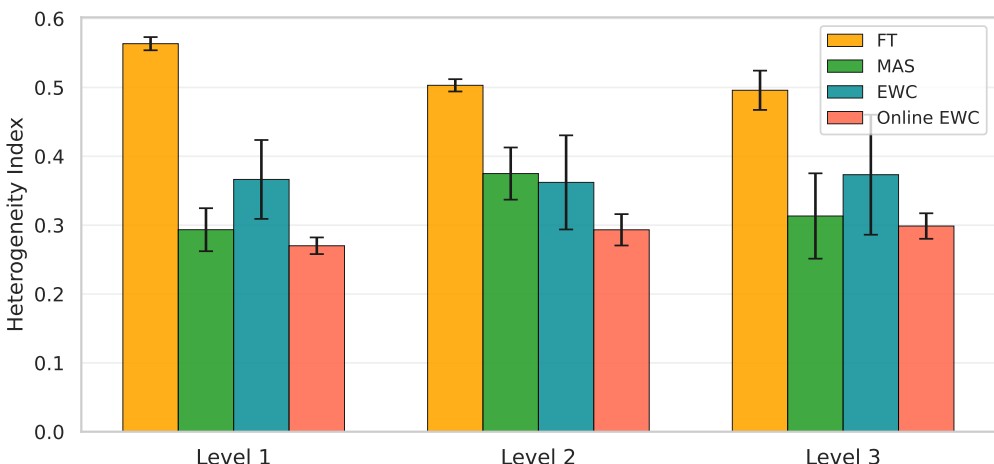

*Figure 17.* **Role Specialisation.** Heterogeneity index $\mathcal{H}$ for each method across difficulty levels. Higher values indicate more distinct roles between agents.

is asymmetric step costs: in many layouts, loading the pot with 3 onions takes more steps than a single plate-and-deliver trip, making the chef the throughput bottleneck. Generalization also suffers as agents struggle to transfer knowledge when their roles change across tasks, since skills learned in one role do not apply to the other. This role-switching dynamic further exacerbates forward transfer challenges in continual learning.

### L.3. Coordination and Role Specialization

The CL metrics in Section 5.1 track soup delivery, but in a cooperative setting a stable score can mask changes in how agents coordinate. A team might keep delivering soup while reorganizing who performs which subtask, or while collapsing from a specialized division of labor into redundant behavior. To capture this, we introduce two coordination-focused measurements that reuse our existing evaluation pipeline.

**Role Specialization.** We quantify how differently the agents behave through a heterogeneity index $\mathcal{H}$, the mean pairwise Jensen–Shannon divergence (JSD) between agent action distributions, averaged over all tasks in the sequence:

$$\mathcal{H} = \frac{1}{N} \sum_{i=1}^{N} \frac{2}{M(M-1)} \sum_{a<b} \text{JSD}\left(\pi_i^a \,\|\, \pi_i^b\right), \tag{12}$$

where $M$ is the number of agents and $\pi_i^a$ the action distribution of agent $a$ on task $i$. A value of $\mathcal{H} = 0$ means the agents act identically (fully homogeneous), while $\mathcal{H} = 1$ means their action distributions do not overlap (fully specialised, for instance one agent cooking soup while the other plates and delivers).

Figure 17 displays $\mathcal{H}$ for each method. FT shows the highest specialisation, as it adapts freely to each task without any constraint on the shared backbone. Among regularization-based methods, EWC retains a slightly stronger role differentiation, whereas Online EWC and MAS settle into more homogeneous behavior. We find that heterogeneity does not depend on the difficulty level.

*Table 13.* **Coordination Forgetting.** CF substitutes the heterogeneity index for the soup score in Eq. 3. $\Delta = \text{CF} - \mathcal{F}$ measures how much more coordination degrades than performance, where positive values indicate coordination is lost before the score drops.

| Method | Level 1 | | | Level 2 | | | Level 3 | | |
|--------|------|------|------|------|------|------|------|------|------|
| | CF | $\mathcal{F}$ | $\Delta$ | CF | $\mathcal{F}$ | $\Delta$ | CF | $\mathcal{F}$ | $\Delta$ |
| EWC | 0.58 | 0.01 | +0.57 | 0.33 | 0.03 | +0.30 | 0.67 | 0.09 | +0.58 |
| Online EWC | 0.28 | 0.06 | +0.22 | 0.29 | 0.10 | +0.19 | 0.25 | 0.14 | +0.11 |
| MAS | 0.31 | 0.30 | +0.01 | 0.14 | 0.36 | −0.22 | 0.45 | 0.45 | 0.00 |
| FT | 0.00 | 0.95 | −0.95 | 0.02 | 0.94 | −0.92 | 0.00 | 0.95 | −0.95 |

**Coordination Forgetting.** The heterogeneity index lets us ask a sharper question, namely, whether coordination structure degrades faster than raw performance. We define coordination forgetting (CF) by substituting $\mathcal{H}$ for the soup score $s_i$ in the forgetting metric (Eq. 3), then compare it against the standard performance forgetting $\mathcal{F}$ through $\Delta = \text{CF} - \mathcal{F}$. A positive $\Delta$ means the cooperative strategy is lost before the score drops.

Table 13 shows that EWC and Online EWC have strongly positive $\Delta$ across all levels. Their scores hold, but their coordination patterns shift underneath. At Level 1, Online EWC forgets little in performance ($\mathcal{F} = 0.06$), yet its role structure degrades almost five times as much CF = 0.28). FT sits at the opposite extreme, where performance and coordination collapse together, leaving $\Delta$ strongly negative and nothing coordination-specific to observe. The larger $\Delta$ for EWC than Online EWC likely stems from how each accumulates its penalty. Since soup delivery is largely invariant to which agent fills which role, many configurations preserve the score while differing in coordination. EWC's cumulative Fisher anchors performance but pulls the shared trunk toward a compromise averaged over all past tasks, washing out their task-specific role structure. Online EWC's decay keeps it closer to the recent coordination pattern, so its performance and coordination stay more aligned.

Together, these results show that agents can keep delivering soup, i.e., maintaining good performance, while their cooperative strategy quietly changes underneath. Standard performance-based CL metrics don't capture this. These metrics we introduced can be used as auxiliary means to investigate this phenomenon in continual MARL.

## M. Continual Partner Adaptation

MEAL enables the generation of diverse partner policies, allowing continual learning methods to be evaluated not only across layouts but also across sequences of partners, e.g., $\mathcal{T} = (\pi_p^0, \ldots, \pi_p^L)$, where $L$ is the sequence length. To this end, we aim to generate partner policy sequences that are maximally diverse in their behaviour. We shall first describe the partners MEAL provides, what order we use for our CL sequence, and finally discuss the results.

### M.1. Partner Description

As described in Section 4.3, we use (i) hardcoded strategies (random, static), (ii) planning-based agents (onion-only, plate-only, and a human-like planner with stochastic task selection), and (iii) populations trained with best-response diversity (BRDiv, Rahman et al., 2023), which maximizes self-play performance while minimizing cross-play compatibility.

BRDiv populations in particular yield highly incompatible strategies. Coordinating with a new BRDiv partner typically requires learning behaviours that differ substantially from those seen before, making them a strong testbed for continual adaptation. In our experiments, we train BRDiv populations with a size of three, a cross-play weight of $1.0$, in $64$ parallel environments, using simple MLP policies.

Planning-based agents follow fixed strategies that learned policies rarely adopt. The onion-only agent collects onions and, with probability $p_{\text{onion-counter}} = 0.1$, places them on counters instead of pots. The plate-only agent collects plates and delivers dishes. With probability $p_{\text{plate-counter}} = 0.1$ it places the plate on a counter instead of plating a soup. The human-like planner follows simple heuristics: it prioritizes filling pots, but if no pot is free, it collects a plate and delivers a soup. With probability $0.1$, it may place either onions or plates on counters instead.

For our experiments, we fix the partner schedule as follows: the ego agent first encounters the three BRDiv partners, then the human-like planner, and finally the onion-only, plate-only, random, and static agents, for eight partners in total. Each partner

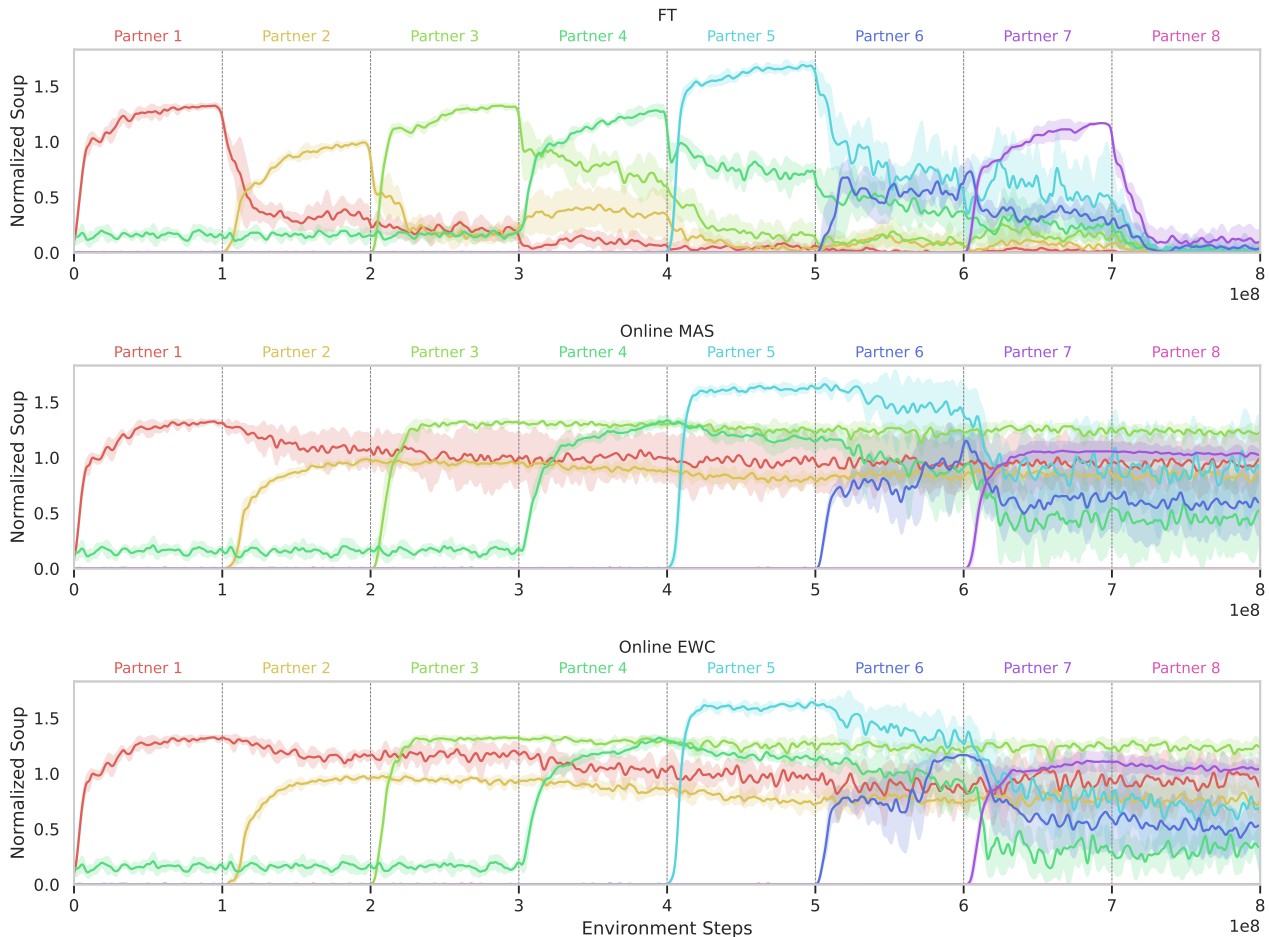

*Figure 18.* Per-task evaluation curves on the *Coordination Ring* layout as the ego agent adapts to eight diverse partners in sequence.

is held fixed while the ego agent trains to collaborate with it for $10^8$ steps. We run this protocol on four widely-adopted hand-designed Overcooked layouts from Carroll et al. (2019) (*Cramped Room*, *Asymmetric Advantages*, *Coordination Ring*, *Counter Circuit*) rather than the procedurally generated kitchens of the *Meal Generator* (Section 4.1), since these layouts are known to be sensitive to partner behaviour (Ruhdorfer et al., 2025b).

### M.2. Evaluation Curves

Figure 18 shows the per-task evaluation curves on Coordination Ring. All three methods learn each partner well within its own training window, reaching comparable peak scores, so the methods differ in retention rather than plasticity. Under FT, each partner's score collapses almost immediately once the next partner begins, with only a short tail of residual coordination. Online MAS and Online EWC retain far more: after a partner is learned, its score stays elevated through much of the remaining sequence, with the earliest partners holding near or above a normalized score of one long after their training window ends. Retention weakens somewhat over the final partners, where curves grow noisier and partly decline, but both online methods clearly preserve more cross-partner coordination than FT.

## N. Reward Settings

By default, Overcooked agents receive *dense shared* rewards. We compare this with an *individual* reward mode, where each agent is rewarded solely for its own actions, often leading to greedier behavior and weaker coordination (Perolat et al., 2017; Foerster et al., 2017; Hughes et al., 2018). We also evaluate the *sparse shared* reward setting described in Section B, assessing all three using EWC on Level 1.

*Table 14.* Effect of reward settings on EWC over Level-1 sequences.

| Rewards | $\mathcal{S}\uparrow$ | $\mathcal{F}\downarrow$ | $\mathcal{FT}\uparrow$ |
|---|---|---|---|
| Dense Shared | **0.90** $\pm$ **0.04** | **0.01** $\pm$ **0.01** | **0.20** $\pm$ **0.08** |
| Dense Individual | 0.84 $\pm$ 0.08 | 0.08 $\pm$ 0.07 | 0.12 $\pm$ 0.06 |
| Sparse Shared | 0.19 $\pm$ 0.04 | 0.02 $\pm$ 0.01 | $-0.79$ $\pm$ 0.10 |

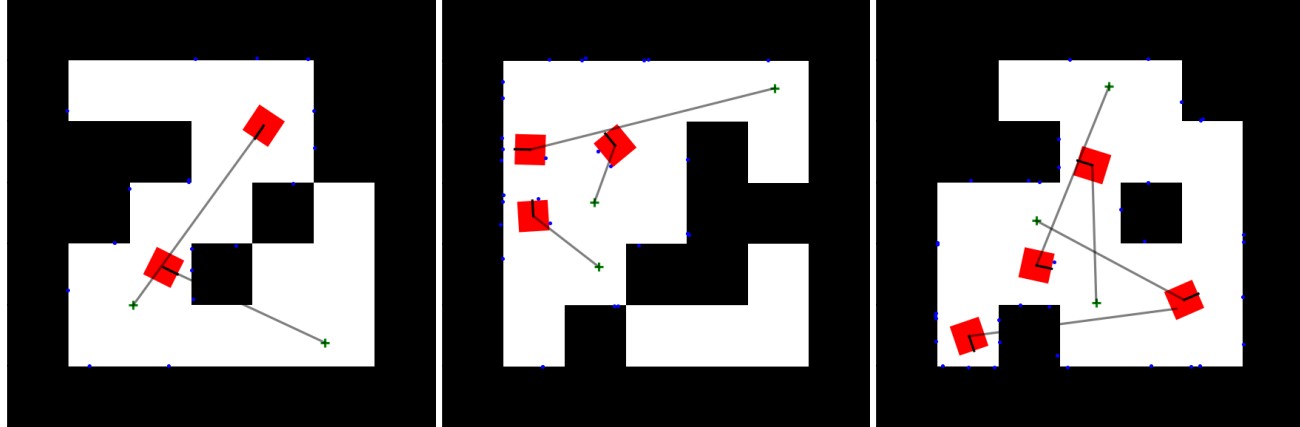

*Figure 19.* Example **JaxNav** environments on a $7\times7$ grid with 2–4 agents. Dark cells denote walls/obstacles, and light cells denote free space. Agents are depicted as red squares with orientation markers. Goal locations are marked with crosses. Thin lines show the direct vector from each agent to its goal. Blue dots visualize lidar sensor returns.

As shown in Table 14, dense shared rewards lead to the best performance, although on some tasks, the individual reward setting allows the agents to find a better global solution due to the inherent competitiveness. Agents are motivated to use different onion piles and pots to maximize their own rewards, which often leads to a more efficient solution. However, this competitive drive occasionally prevents them from converging on a stable solution. The sparse reward setting grants rewards only for successful deliveries. Without a targeted exploration mechanism, agents are unlikely to discover a full delivery sequence through random actions even on Level 1 layouts, leading to worse performance.

## O. Extending MEAL Beyond Overcooked

While this paper centers MEAL around Overcooked, the continual learning framework is not restricted to a single domain. To demonstrate this, we incorporate three additional cooperative JAX-based environments: JaxNav (Rutherford et al., 2024a), SMAX (Rutherford et al., 2024b), and MPE SimpleSpread (Lowe et al., 2017; Mordatch & Abbeel, 2018). Each introduces distinct coordination challenges, observation modalities, and action spaces, while preserving the high-throughput, GPU-accelerated training pipeline that defines MEAL.

### O.1. JaxNav

JaxNav is a navigation-based continuous multi-agent environment, introducing several challenges: agents must reach their assigned goal locations relying on local lidar observations while avoiding collisions with walls and each other. Some layouts contain narrow bottlenecks where two agents can not pass simultaneously, requiring one to explicitly give way to the other(s).

To create continual learning task sequences in JaxNav, we rely on its built-in randomized layout generator. For our experiments, we use $7\times7$ grids with an obstacle fill ratio of $0.3$. Unlike Overcooked, where we derive continual learning metrics from the normalized soup delivery score, JaxNav allows us to compute these metrics directly from the environment's raw returns. We keep all training settings identical to Overcooked, including the MLP encoder, PPO algorithm, Adam optimizer, hyperparameters, and evaluation schedule. We run 20-task sequences over 5 seeds and evaluate Online EWC with 2, 3, and 4 agents. Figure 19 depicts example environments used in our experiments.

Figure 20 shows a slight downward trend in performance when increasing the number of agents. Importantly, the increase

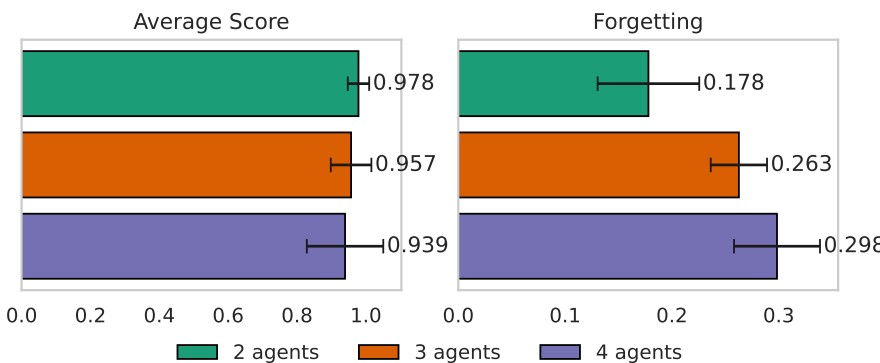

*Figure 20.* Online EWC performance on 20-task JAXNAV sequences. Forgetting increases steadily with additional agents, while the average score degrades more mildly.

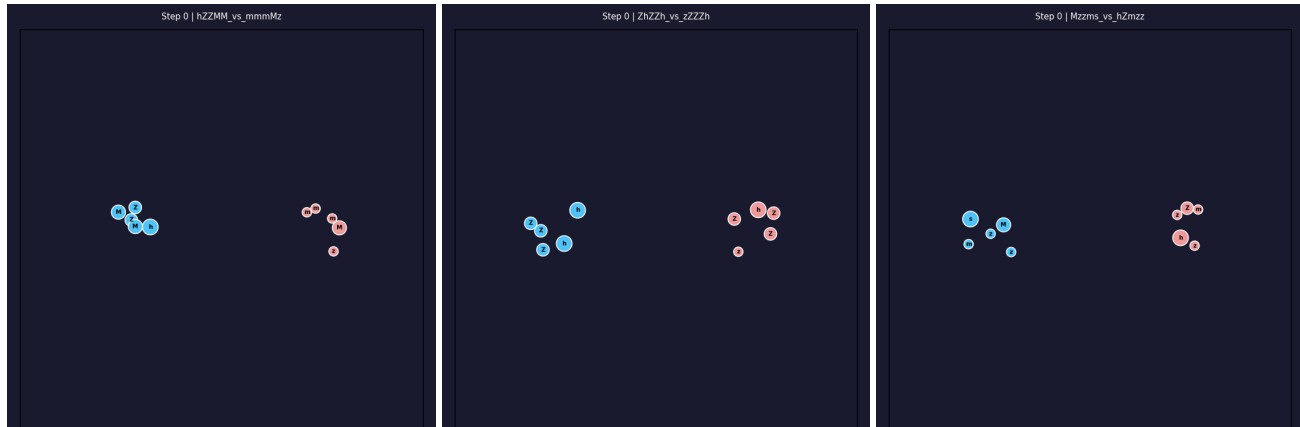

*Figure 21.* Three example **SMAX** tasks from a single sequence. Blue units (left) form the ally team, pink units (right) form the script-controlled enemy team. Each circle is a unit, labeled by its type, with size reflecting the heterogeneous unit classes (varying health, range, and movement). Each task draws a fresh ally and enemy composition from six unit types, as encoded in the matchup names (e.g., hZZMM_vs_mmmMz), while team sizes stay fixed. The shifting compositions across tasks are the source of non-stationarity that the continual learner must adapt to.

in forgetting is more notable, meaning that tasks with more agents are harder to both learn and remember. This once again reflects a key point of our work: continual learning becomes more difficult as the number of interacting agents increases. The addition of JAXNAV shows that MEAL naturally extends to other JAX-based environments while preserving its high-throughput training pipeline.

### O.2. SMAX

SMAX (Rutherford et al., 2024b) is the JAX reimplementation of the StarCraft Multi-Agent Challenge (SMAC) (Samvelyan et al., 2019), a widely used cooperative MARL benchmark. Each agent controls a single unit in a team-versus-team micromanagement scenario, where allied units must coordinate movement and focus their fire to defeat a script-controlled enemy team. Unlike Overcooked and JAXNAV, SMAX scales naturally to larger teams and supports heterogeneous units that differ in health, range, and movement. This makes it well-suited for studying continual coordination as both the number and the diversity of agents grow, a regime that is difficult to probe in Overcooked.

We form task sequences by randomly sampling ally and enemy unit compositions from six unit types per task, keeping team sizes fixed. With teams of, e.g., five, this yields $6^5 \cdot 6^5 \approx 60M$ possible matchups. Since random compositions can produce highly unbalanced matchups where the original winrate metric becomes uninformative, we instead measure performance as the fraction of enemy units eliminated.

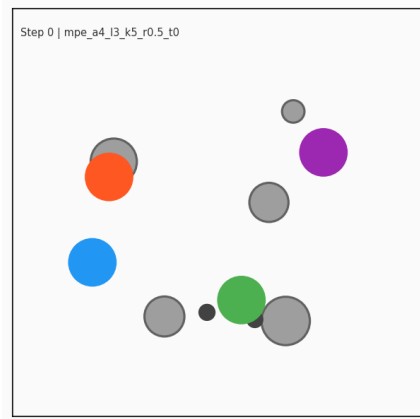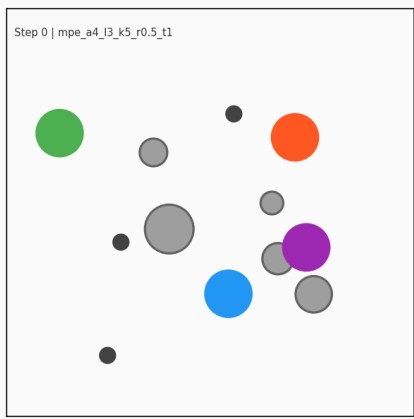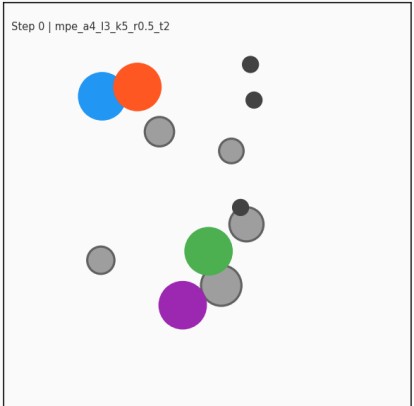

*Figure 22.* Three example MPE SIMPLESPREAD tasks from a single sequence (`a4_l3_k5`: four agents, three landmarks, five obstacles). Colored circles are agents, small dark circles are landmarks to be covered, and gray circles are obstacles of varying radii that block movement. Agent and landmark positions are randomized each episode, while obstacle positions and radii are fixed within a task but differ across tasks, producing the distributional shift the continual learner must adapt to.

### O.3. MPE SimpleSpread

MPE SIMPLESPREAD (Lowe et al., 2017; Mordatch & Abbeel, 2018) is a cooperative navigation task from the Multi-Agent Particle Environments. $N$ agents must spread out to cover $N$ landmarks while avoiding collisions, receiving a shared reward based on the distance from each landmark to its nearest agent. The continuous state space and the implicit need to divide landmarks among agents without communication make it a nice testbed for emergent role allocation under continual non-stationarity.

To make coordination harder, we introduce obstacles that inhibit agent movement (see Figure 22). Agent and landmark positions are randomized per episode as usual, but we fix the obstacle positions and radii per task in the sequence. We measure performance as the fraction of landmarks covered.

### O.4. Scaling the Number of Agents

To test whether the agent-count trends from Overcooked and JaxNav hold in other domains, we evaluate Online EWC on MPE SimpleSpread and SMAX across varying team sizes (Table 15). In both environments, average performance degrades as agents are added, consistent with our earlier findings that continual learning grows harder with more interacting agents.

Forgetting behaves differently across the two. In SMAX, forgetting rises with team size, mirroring Overcooked. In MPE, it stays roughly flat or even edges down. A likely explanation is that with more agents, peak performance in MPE is already low, so there is less to forget in the first place.

*Table 15.* Online EWC across varying agent counts on MPE SimpleSpread and SMAX, reported as average performance $\mathcal{A}$ and forgetting $\mathcal{F}$. SMAX is evaluated from five agents upward, since its team-versus-team scenarios require larger teams.

| Agents | MPE SimpleSpread | | SMAX | |
|---|---|---|---|---|
| | $\mathcal{A}\uparrow$ | $\mathcal{F}\downarrow$ | $\mathcal{A}\uparrow$ | $\mathcal{F}\downarrow$ |
| 3 | 0.52 | 0.05 | – | – |
| 4 | 0.39 | 0.05 | – | – |
| 5 | 0.27 | 0.05 | 0.57 | 1.33 |
| 6 | 0.21 | 0.04 | 0.47 | 1.67 |
| 7 | 0.19 | 0.04 | 0.42 | 1.60 |
| 8 | 0.18 | 0.04 | 0.39 | 1.71 |

## P. Limitations

While MEAL provides a scalable and diverse testbed for CMARL, several limitations remain. First, Overcooked is restricted to discrete action spaces, limiting its applicability. Second, while layout diversity is high, the domain itself is narrow. Overcooked dynamics do not capture the full complexity of real-world multi-agent interactions. Third, our benchmark only evaluates task-incremental learning by changing layouts, partners, and non-stationarity factors. Future work could extend MEAL to other CL settings. Finally, although MEAL provides substantial layout diversity, the underlying task structure remains fixed. Future work could extend the benchmark with additional recipe types, ingredients, or preparation mechanics to introduce broader semantic variation across tasks.

## Q. Training and Evaluation Curves

This section presents additional training and evaluation curves, providing a more detailed view of per-task evaluation and transfer. Figure 25 depicts the per-task evaluation curves of Level 1. Figure 26 illustrates forward transfer.

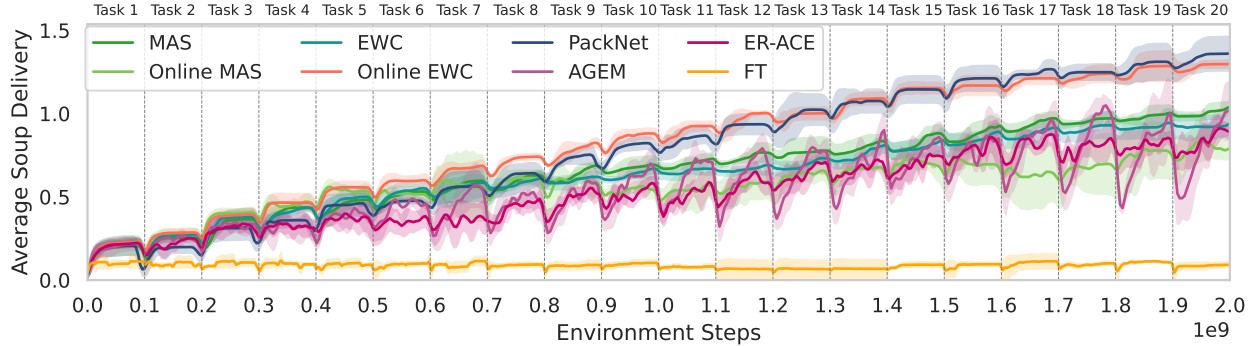

*Figure 23.* **Average Soup Delivery** curves of CL baselines combined with IPPO on Level 2.

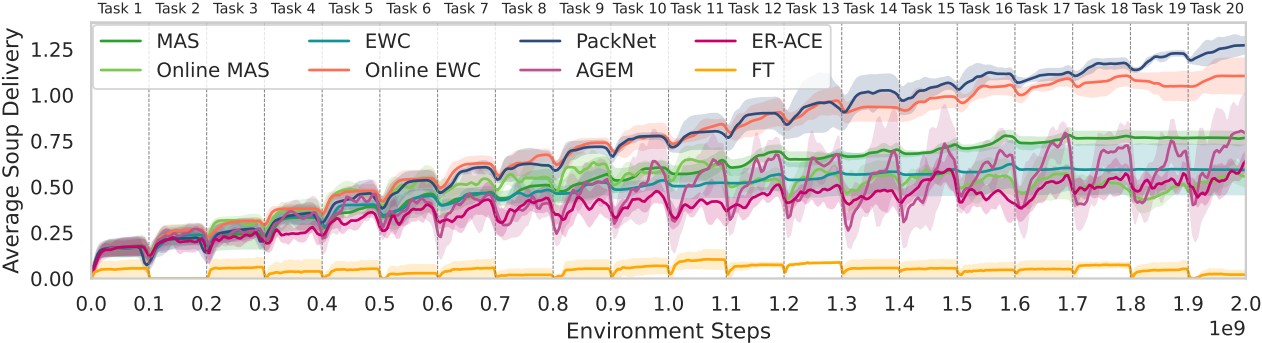

*Figure 24.* **Average Soup Delivery** curves of CL baselines combined with IPPO on Level 3.

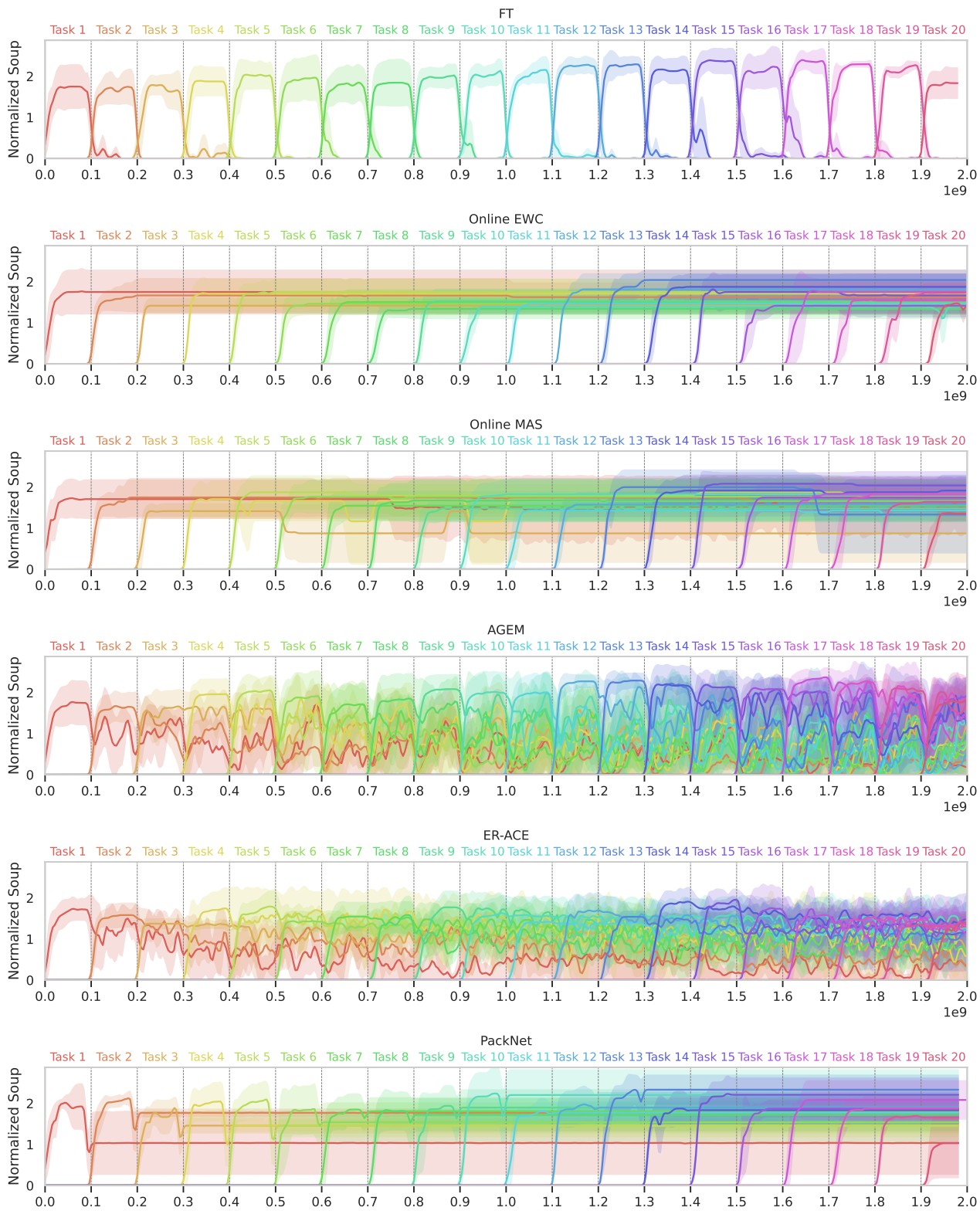

*Figure 25.* The **evaluation curves** of Level 1 illustrate the extent of forgetting across tasks. FT suffers from clear catastrophic forgetting: once the agent transitions to a new task, performance on the previous task collapses immediately. Online EWC displays near-perfect retention.

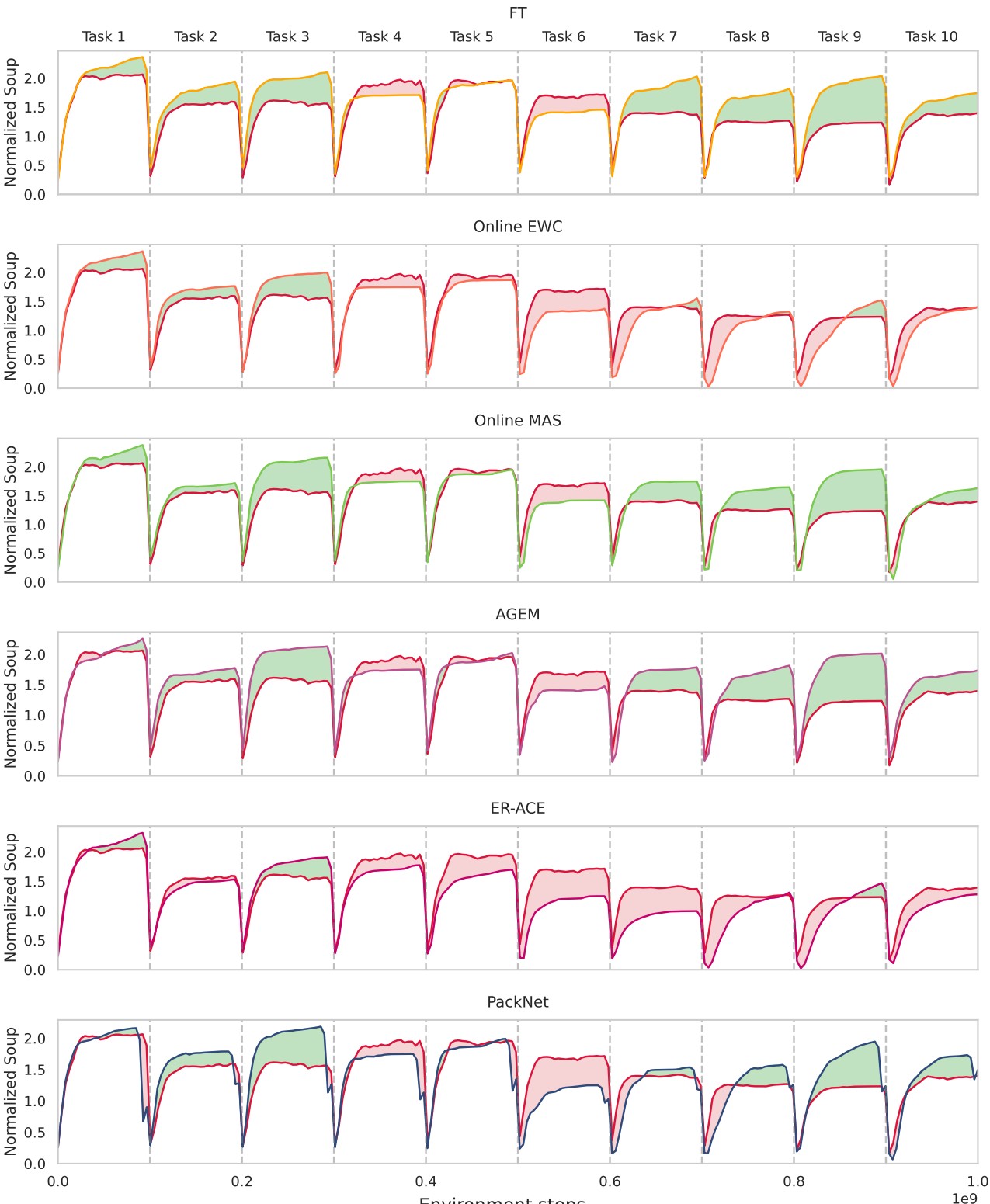

*Figure 26.* **Forward transfer** on Level 2. The green shaded areas depict positive transfer compared to the IPPO baseline, and the red shaded areas show negative transfer.

