# OpenReview forum: "MEAL: A Benchmark for Continual Multi-Agent Reinforcement Learning"
_ICML.cc/2026/Conference — ICML 2026 regular_

### Official Review · Reviewer_P2ez · 2026-03-09

**Soundness:** 2
**Presentation:** 3
**Significance:** 3
**Originality:** 2
**Overall Recommendation:** 3
**Confidence:** 4

**Summary:**

This paper introduces MEAL, a benchmark designed for continual multi-agent reinforcement learning (CMARL). MEAL adopts a JAX-based implementation to enable hardware-accelerated continual experiments. Specifically focused on the existing cooperative MARL environment Overcooked environment, MEAL provides discrete task sequences equipped with a chosen level of difficulty and evaluates variants of existing continual learning (CL) algorithms built upon IPPO based on CL metrics. Experiments show that the number of agents as well as the length of the task sequence affect the difficulty of continual MARL tasks.

**Compliance With Llm Reviewing Policy:**

Affirmed.

**Final Justification:**

I have read the authors' reply carefully. The authors mention that "we provide the results for up to 8 agents (see below). For the time being, we have not evaluated with a higher number. However, it is trivial to do so." I strongly believe that providing the result with more agents is necessary because the authors claim that their benchmark is tailored for multi-agent settings. Also, the current version of this paper considers curriculum experiment as an analytical purpose, but I believe including extensive results is necessary to cover continual learning in-depth.

Despite these points, I have raised my score from 2 to 3 since the authors addressed many of my concerns during the rebuttal. I strongly recommend the authors to include the points mentioned above in the future.

**Key Questions For Authors:**

See the Weaknesses above.

- In Conclusion, what does this mean and how this claim is supported in this paper? “Individual rewards weaken continual coordination and induce negative transfer.”

**Limitations:**

The authors did not include Impact Statement in the paper, but discuss some limitations in Appendix N.

**Strengths And Weaknesses:**

Strengths
- Continual RL is an important domain in RL yet the benchmark is limited, so implementing the benchmark is meaningful.
- MEAL supports the increase of the task sequence length up to 100 which enables deeper analysis on continual RL/MARL.

Weaknesses
1. On the contribution for ‘multi-agent’ CRL benchmark
- To demonstrate the novelty of the proposed benchmark specially tailored for ‘multi-agent’ CRL, MEAL should allow tools for deeper analysis on special issues of MARL such as non-stationarity, credit assignment, communication, heterogeneity of agents, and so on. To support these, the core requirement is to allow the number of agents increase a lot, in addition to the task sequence length. However, in Section 5.3, experiments in Overcooked provides the setting up to 4 agents. I’m not saying that 4 agents setting is trivial or the authors should increase the number of agents in this environment because Overcooked environment itself may not be adequate to encompass a lot of agents such as more than 10. For example, like in JaxMARL, it would be meaningful to design a benchmark upon SMAC/SMAX which incorporates the increase of the number of agents more than Overcooked, as well as allowing heterogeneity of agents.
This seems essential because this is a benchmark paper, but other than Section 5.3, it follows the similar logic of existing CRL work such as Continual World (Wołczyk et al., 2021), which may not position this paper specially tailored for ‘multi-agent’ CRL.
- Proposing another metric related to special issues of MARL such as non-stationarity, credit assignment, communication, heterogeneity of agents, and asynchrony of action executions, in the context of continual learning can strengthen the novelty of this paper specially tailored for ‘multi-agent’ CRL.

2. Strengthening the analysis on the ‘curriculum’ setting further improve the contribution of this paper, since in general, we cannot always guarantee that the given task sequence has constant level of difficulty.

3. Some of the key experiments should be clarified.
- What is the chosen value of lambda in Eq.2? How is the value is properly chosen?
- In Figure 5, why the performance of EWC increases when we ablate task identity inputs?

Minor
- Erden et al. year is missed. (Line 30)

---

> ### Author Rebuttal · Authors · 2026-03-29
>
> **MARL-Specific Analysis.** We agree that the MARL side of MEAL should be more deeply investigated. We believe the most meaningful aspect is heterogeneity and coordination across tasks. This means to determine whether the way agents cooperate is retained across tasks, not just whether the score holds up. We introduce two types of analysis:
>
> **(1) Role Specialisation**. We measure a heterogeneity index $H \in [0,1]$ defined as the mean pairwise Jensen-Shannon divergence between agents' action distributions, averaged over all tasks. This can also be thought of as a *Role Specialisation Score*. $H = 0$ means agents take identical actions (fully homogeneous) and $H = 1$ means their action distributions are non-overlapping (completely different roles, e.g., one cooks soup, the other plates and delivers).
>
> |Method|Level 1|Level 2|Level 3|
> |---|---|---|---|
> |FT|0.564|0.503|0.496|
> |EWC|0.366|0.362|0.373|
> |MAS|0.293|0.375|0.313|
> |L2|0.301|0.255|0.241|
> |Online EWC|0.270|0.293|0.299|
>
> FT has the highest heterogeneity because it freely adapts to each task. Among methods that have a low forgetting on MEAL, EWC maintains the strongest role differentiation, while Online EWC and L2 produce more homogeneous behavior.
>
> **(2) Coordination Forgetting**. The heterogeneity index lets us ask a deeper question: does coordination structure degrade more than performance does? To answer this, we compute coordination forgetting (**CF**). We use Eq. 3 in the paper by substituting $H$ for the original soup score.
>
> We can then compare this value with the normal performance forgetting **F**. The $\Delta = \text{CF} - \text{PF}$ captures whether coordination structure degrades proportionally more than performance. A positive $\Delta$ means coordination is lost before performance drops.
>
> |Method|L1 CF|L1 F|L1 Δ|L2 CF|L2 F|L2 Δ|L3 CF|L3 F|L3 Δ|
> |---|---|---|---|---|---|---|---|---|---|
> |EWC|0.58|0.01|+0.57|0.33|0.03|+0.30|0.67|0.09|+0.58|
> |Online EWC|0.28|0.06|+0.22|0.29|0.10|+0.19|0.25|0.14|+0.11|
> |L2|0.36|0.02|+0.34|0.53|0.06|+0.47|0.50|0.10|+0.40|
> |MAS|0.31|0.30|+0.01|0.14|0.36|−0.22|0.45|0.45|0.00|
> |FT|0.00|0.95|−0.95|0.02|0.94|−0.92|0.00|0.95|−0.95|
>
> EWC and Online EWC show strongly positive Δ across all levels: scores are maintained but coordination patterns change. Online EWC at Level 1 has F = 0.06, yet CF = 0.28. The role structure degrades nearly 5× more than performance. FT is the opposite: everything collapses together, so Δ is strongly negative, and there is nothing coordination-specific to observe.
>
> Key takeaway: Decent performance can be maintained even when the cooperative strategy inadvertently changes.
>
> **Few Agents**. Overcooked and JaxNav are indeed limited in how many agents they can meaningfully support. In response to this and other reviewers, we have incorporated two additional environments: MPE and, as you suggested, SMAX. Both naturally scale to higher agent counts. While in MEAL, we designed a heterogeneous setting *artificially* (Appendix H) by assigning agents to designated roles, in SMAX, heterogeneity arises implicitly. Please see our response to **Reviewer bbE3 (W1)** for the results and discussion. We believe these 4 environments already cover a much broader range of opportunities in CMARL.
>
> **Curriculum Analysis**. Our curriculum experiment mainly serves an analytical purpose: validating that our difficulty levels represent genuinely harder learning problems, not just structural constraints (e.g., fewer onions/pots capping the soup score). The results rule this out. Agents pre-trained on easier levels reach substantially higher Level 3 performance than agents trained on Level 3 from scratch, confirming that skills transfer and that there is room for improvement. Studying different orderings (reverse, random) is interesting future work that MEAL's procedural generation makes easy to set up.
>
> **λ value**. Thank you for pointing this out. Indeed, we had omitted this value and will include it. We use $λ=2$. The exponential weight $w(t) = exp(-λ·(t-τᵢ)/(T-τᵢ))$ penalizes early forgetting more than late. $λ=2$ means that at the end of the sequence, $w = exp(-2) ≈ 0.14$, so late forgetting still contributes but at $\sim7×$ less than immediate forgetting. At the halfway point, $w = exp(-1) ≈ 0.37$, so most of the training horizon contributes meaningfully. In response to **Reviewer bbE3 (W3)**, we ablated $λ$ across [0, 8] to show its sensitivity.
>
> **EWC Task ID**. Since the default setting used multiple output heads and the correct output head already identified the task, we believe the additional task identity input became redundant. The extra input dimensions add a small amount of noise to the shared representation that EWC then has to protect that carries no useful information and complicates optimization.
>
> **Individual Rewards**. This refers to our reward setting comparison in Appendix L (Table 12), where individual rewards yielded lower scores, higher forgetting, and worse transfer.

---

> > ### Author Rebuttal · Reviewer_P2ez · 2026-04-01
> >
> > Thanks for the reply. I have additional questions.
> >
> > 1. Regarding the SMAX environment, could the authors provide a detailed breakdown of the six unit types? Also, can the number of agents be increased larger than 5?
> >
> > 2. For the MPE and SMAX benchmarks, would the authors consider including FT results? This would ensure consistency with the metrics reported for the Overcooked environment.
> >
> > 3. When coordination patterns change in EWC/Online EWC, is this shift driven by the algorithm’s proactive adaptation to an altered setting (a positive outcome), or is it a result of catastrophic forgetting (a negative outcome)? Could the authors provide qualitative analysis explaining the "snapshot" of that moment?
> >
> > 4. Is the "role specialization score" (computed via JS) only well-defined for environments with homogeneous agents? Or can we well-define it with heterogeneous agents?

---

> > > ### Author Response · Authors · 2026-04-05
> > >
> > > Thank you for the thoughtful follow-up questions.
> > >
> > > > Regarding the SMAX environment, could the authors provide a detailed breakdown of the six unit types?
> > >
> > > Yes, of course. We use the same units as SMAX and the original SMAC:
> > >
> > > | Abbrev. | Unit       | Race    |
> > > |-----------|-----------|---------|
> > > | m         | Marine    | Terran  |
> > > | M         | Marauder  | Terran  |
> > > | s         | Stalker   | Protoss |
> > > | Z         | Zealot    | Protoss |
> > > | z         | Zergling  | Zerg    |
> > > | h         | Hydralisk | Zerg    |
> > >
> > > > Also, can the number of agents be increased larger than 5?
> > >
> > > Indeed, we provide the results for up to 8 agents (see below). For the time being, we have not evaluated with a higher number. However, it is trivial to do so. There is no practical limitation on how large the team size can be, apart from the map size itself. Our default map size can reasonably accommodate 15-20 agents per team. If one wishes to run more, it would be practical to increase the map size. When we compose the team from 6 unit types, then with N agents, we'd get N independent draws, and the same unit type can appear multiple times.
> > >
> > >
> > > > For the MPE and SMAX benchmarks, would the authors consider including FT results? This would ensure consistency with the metrics reported for the Overcooked environment.
> > >
> > > |Agents|MPE A↑|MPE F↓|MPE FT↑|SMAX A↑|SMAX F↓|SMAX FT↑|
> > > |------|------|------|------|------|------|------|
> > > |3|**0.518**|0.051|0.537|-|-|-|
> > > |4|0.392|0.051|0.576|-|-|-|
> > > |5|0.267|0.045|0.818|**0.570**|**1.330**|-0.717|
> > > |6|0.210|0.043|1.302|0.472|1.666|-0.653|
> > > |7|0.189|**0.043**|2.087|0.422|1.599|**-0.420**|
> > > |8|0.180|0.043|**2.408**|0.485|1.514|-0.507|
> > >
> > > FT increases with more agents, not necessarily because the continual learner gets better, but because the single-task baseline gets worse. Learning the task from scratch becomes a bigger struggle with larger teams. This is where the continual learner shines, as prior experience is more valuable. The negative FT in SMAX suggests that learned unit compositions interfere with new ones.
> > >
> > > > When coordination patterns change in EWC/Online EWC, is this shift driven by the algorithm’s proactive adaptation to an altered setting (a positive outcome), or is it a result of catastrophic forgetting (a negative outcome)? Could the authors provide qualitative analysis explaining the "snapshot" of that moment?
> > >
> > > For regularization-based methods like EWC and Online EWC, this shift can only be unintentional (**negative**). These methods try to anchor important weights to preserve past task performance. They have no mechanism to improve performance nor coordination on tasks that have already been trained on. So when we observe coordination patterns changing on a previously learned task, it must be due to drift in the shared backbone that the regularization failed to prevent fully.
> > >
> > > The question is then: if coordination is degrading, why does performance hold up? At least on the lower Overcooked levels, that is the case. We observed some gameplay recordings stored during evaluation and noticed a plausible explanation. Level 1 (and 2) layouts are smaller, and agents are less likely to get separated in different parts of the map (Figure 2 in the paper). Therefore, homogeneous behavior still works. Two agents independently running the same cook-deliver cycle can still produce a reasonable number of soups, and the scores stay high. On Level 3, specialization is more required because a single agent likely cannot access all the components. Online EWC's Δ reflects this, decreasing 0.22 (Level 1) $\rightarrow$ 0.19 (Level 2) $\rightarrow$ 0.11 (Level 3). This is what makes comparing $CF$ with $F$ meaningful.
> > >
> > > > Is the "role specialization score" (computed via JS) only well-defined for environments with homogeneous agents? Or can we well-define it with heterogeneous agents?
> > >
> > > That is a great question. Mathematically: yes. Practically: yes, but the interpretation changes if different actions are masked out between agents, i.e., they are restricted from executing them. This means, in SMAX (and most heterogeneous MARL environments) it works the same way, since every agent type can perform the same actions. However, on the chef/waiter setting we created for overcooked, the interpretation is different. Agents are already heterogeneous by design because they cannot perform the same actions, and the action distributions will naturally differ. The metric is still useful, though, but we should now measure the change of $H$ over time rather than the absolute value. For instance, if H increases from 0.3 to 0.6 across tasks, then agents developed more specialized strategies aside from the "baked-in" heterogeneity. In short, our Coordination Forgetting (CF) metric is identical, but the raw $H$ depends on the setting. We shall measure and report this in the paper for all environments, not only Overcooked.
> > >
> > > We hope to have clarified any remaining points. Thank you for reviewing our work and helping to improve it.

---

### Official Review · Reviewer_bbE3 · 2026-03-11

**Soundness:** 2
**Presentation:** 3
**Significance:** 3
**Originality:** 2
**Overall Recommendation:** 5
**Confidence:** 3

**Summary:**

This paper introduces MEAL (Multi-agent Environments for Adaptive Learning), a JAX-based benchmark for continual multi-agent reinforcement learning (CMARL). The paper studies the intersection of continual learning and cooperative MARL by building on the Overcooked environment with procedural generation and GPU-accelerated training. The benchmark supports sequences of up to 100 tasks on a single GPU in a few hours, which represents a meaningful engineering contribution to the field.

**Compliance With Llm Reviewing Policy:**

Affirmed.

**Final Justification:**

The rebuttal addresses most of my concerns and strengthens the paper significantly, particularly through additional experiments and broader evaluation.
While the coverage of more recent continual learning baselines remains somewhat limited, this is a minor weakness that does not undermine the overall contribution.
I increase my score accordingly.

**Key Questions For Authors:**

The main finding of this study highlights the importance of long task sequences for revealing meaningful differences between continual learning methods in multi-agent settings — a finding that is well-supported and practically valuable. However, the paper's positioning as a general CMARL benchmark is undermined by its reliance on a single domain, an outdated baseline set, and underexplored MARL algorithm diversity. The authors should be encouraged to address these weaknesses and limitations of the paper.

**Limitations:**

The evaluated CL methods (EWC, Online EWC, L2, MAS, AGEM, Fine-Tuning) are all from 2017–2018. The field has advanced considerably. Methods like PackNet, DER++, ER-ACE, CLONEX, and more recent plasticity-preserving methods (e.g., ReDo, Shrink-and-Perturb) are conspicuously absent. For a 2026 submission, benchmarking only against methods that are nearly a decade old is difficult to defend, particularly when the paper claims to "provide headroom for future methods."

**Strengths And Weaknesses:**

**Strengths**:

**S1** The paper correctly identifies that continual MARL is largely unexplored territory. The combination of properties in Table 1 — GPU-accelerated, multi-agent, continual, procedurally generated — is impressive.

**S2** Reducing 100-task training to ~4 hours on a single H100 is practically significant and will lower the barrier for researchers without massive compute budgets.

**S3** Section 5.5 introduces a partially observable variant and finds the surprising result that MAPPO underperforms IPPO in the CL setting. This is potentially a significant finding — centralized training may actively harm continual learning — but it receives only one paragraph of analysis and one figure. This deserves much deeper investigation.

**Weaknesses**:

**W1** A benchmark that claims to advance CMARL but is validated on a single cooperative cooking game will struggle to generalize its findings.

**W2** The forgetting metric (Equation 3) uses an exponential decay weighting that "penalizes earlier forgetting more strongly." This is a non-standard and somewhat arbitrary design choice. The authors do not justify why earlier forgetting should be penalized more — in many lifelong learning scenarios, the opposite argument could be made (older tasks matter less). This could distort comparisons with prior work using standard forgetting metrics.

**W3** The forgetting metric's λ parameter (Eq. 2) is never ablated. How sensitive are results to this choice?

**W4** Only IPPO is used as the base MARL algorithm, with MAPPO appearing briefly in Section 5.5. Cooperative MARL has many established algorithms — QMIX, MADDPG, MAPPO variants, HAPPO — and it is unclear whether the CL findings are IPPO-specific or general.

---

> ### Author Rebuttal · Authors · 2026-03-30
>
> **W1 Only Overcooked** We agree. Beyond Overcooked and JaxNav, we now include two additional environments: **MPE SimpleSpread** and **SMAX**. For MPE, we introduce obstacles that inhibit movement, making coordination harder. Agent and landmark positions are randomized per episode, as usual, but we fix the obstacle positions and radii per task in the sequence. We measure performance as the fraction of landmarks covered. For SMAX, we randomly sample enemy and ally unit compositions from 6 unit types per task, keeping team sizes fixed. With teams of size 5, this yields $6^5 x 6^5 \approx60M$ possible matchups. Since this may create very unbalanced matchups, the original winrate metric becomes less informative. Instead, we measure the fraction of enemies eliminated. We evaluate Online EWC across varying agent counts:
>
> |Agents|MPE A↑|MPE F↓|SMAX A↑|SMAX F↓|
> |------|------|------|------|------|
> |3|**0.518**|0.051|-|-|
> |4|0.392|0.051|-|-|
> |5|0.267|0.045|**0.57**|**1.33**|
> |6|0.210|0.043|0.47|1.67|
> |7|0.189|**0.043**|0.42|1.60|
> |8|0.180|0.043|0.39|1.71|
>
> Both confirm our Overcooked and JaxNav findings: performance degrades with more agents. In SMAX, forgetting also increases, which is the opposite in MPE. Likely because with more agents, peak performance is lower to begin with, leaving less to forget. We shall incorporate these results into the main paper.
>
> **W2 Forgetting Metric**. The standard forgetting metric in e.g., Continual World [1] and COOM [2] only measures the drop between peak and final performance, ignoring *when* forgetting occurs. Consider a 10-task sequence where CL methods A and B both learn task 1 equally well. Method A forgets task 1 immediately when training on task 2. Method B retains it through tasks 2–9 and only loses it on task 10. Standard forgetting scores these scenarios identically. We believe this does not tell the whole story. Method B shows good retention for 8 tasks, while method A retained nothing. Our weighted metric captures this distinction.
>
> We don't see why older tasks should matter less, either. The standard CL objective shared by our paper, Continual World, and COOM, is to maximize performance on *all* tasks equally: $A = (1/N) Σ sᵢ(T)$. There is no task-recency discount anywhere in this formulation. A method that forgets task 1 immediately is strictly further from this objective than one that retains it for 8 more tasks, even if both end at the same final score. Our metric captures this: it doesn't deprioritize any task, it deprioritizes *late* forgetting relative to *immediate* forgetting. Besides, setting $λ=0$ recovers unweighted forgetting, so our formulation is strictly more general.
>
> **W3 λ Ablation**. We use $λ=2.0$. We ablate on Level 3, where forgetting is the strongest. Our method ranking largely remains stable across all values. The only swap is between EWC and L2, which differ by $\sim0.01$.
>
> |Method|λ=0.0|λ=0.5|λ=1.0|λ=2.0|λ=4.0|λ=8.0|
> |---|---|---|---|---|---|---|
> |FT|0.948|0.947|0.947|0.947|0.946|0.944|
> |AGEM|0.876|0.872|0.868|0.861|0.847|0.829|
> |MAS|0.598|0.562|0.524|0.450|0.323|0.177|
> |EWC|0.122|0.114|0.106|0.091|0.066|0.043|
> |L2|0.112|0.108|0.104|0.096|0.086|0.077|
> |Online EWC|0.036|0.032|0.028|0.021|0.011|0.004|
> |PackNet|0.005|0.005|0.005|0.005|0.004|0.004|
>
> FT and AGEM are nearly invariant to λ because they forget immediately upon task switch, so the exponential weighting has little to discount. PackNet is similarly stable since it barely forgets at all. In contrast, methods with gradual forgetting (MAS, EWC, Online EWC) are more sensitive, as λ down-weights their slow drift over time.
>
> **W4 Only IPPO**. We have additionally implemented PackNet as a CL baseline, and HAPPO, VDN, and QMIX as MARL algorithms. We also ran MAPPO on the full suite. We haven't run PackNet with other MARL algorithms yet. We use Online versions of EWC and MAS and report the results of Level 1.
>
> |Algorithm|FT A↑|FT F↓|EWC A↑|EWC F↓|MAS A↑|MAS F↓|PackNet A↑|PackNet F↓|
> |---|---|---|---|---|---|---|---|---|
> |IPPO|0.048|0.899|1.612|0.032|1.633|0.082|**1.746**|0.211|
> |MAPPO|0.092|2.010|1.575|0.032|1.581|0.066|–|–|
> |HAPPO|**0.104**|2.075|**1.837**|**0.030**|**1.715**|0.161|–|–|
> |VDN|0.033|1.544|0.201|0.834|0.195|1.220|–|–|
> |QMIX|0.022|**0.144**|0.320|0.067|0.281|**0.041**|–|–|
>
> VDN and QMIX perform poorly, which is consistent with JaxMARL [3], where these methods are already weaker on Overcooked even without CL. Q-value overestimation frequently gets agents stuck in repetitive action cycles.
>
> [1] Wołczyk, Maciej, et al. "Continual world: A robotic benchmark for continual reinforcement learning." Advances in Neural Information Processing Systems 34 (2021).
>
> [2] Tomilin, Tristan, et al. "Coom: A game benchmark for continual reinforcement learning." Advances in Neural Information Processing Systems 36 (2023).
>
> [3] Rutherford, Alexander, et al. "Jaxmarl: Multi-agent rl environments and algorithms in jax." Advances in Neural Information Processing Systems 37 (2024).

---

> > ### Author Rebuttal · Reviewer_bbE3 · 2026-04-03
> >
> > Thank you for the detailed rebuttal. The additional experiments and clarifications address most of my concerns, and I will increase my score.
> > The inclusion of new environments (MPE, SMAX), λ ablation, and evaluation across multiple MARL algorithms significantly strengthen the paper.
> > While the addition of PackNet partially improves baseline diversity, I recommend including stronger replay-based baselines (e.g., DER++, ER-ACE) in the main text, as this would further strengthen the benchmark.

---

> > > ### Author Response · Authors · 2026-04-05
> > >
> > > Thank you very much for your careful review, encouraging feedback, and suggestions to improve our work. We have also included ER-ACE to further expand our baseline coverage. The updated Level 1 results on Overcooked are shown in the table below. To better compare CL methods across MARL algorithms, we transposed the table.
> > >
> > > |Method|IPPO A↑|IPPO F↓|MAPPO A↑|MAPPO F↓|HAPPO A↑|HAPPO F↓|VDN A↑|VDN F↓|QMIX A↑|QMIX F↓|
> > > |---|---|---|---|---|---|---|---|---|---|---|
> > > |FT|0.048|0.899|0.092|2.010|0.104|2.075|0.033|1.544|0.022|0.144|
> > > |EWC|1.612|**0.032**|1.575|**0.032**|**1.837**|**0.030**|**0.201**|**0.834**|**0.320**|0.067|
> > > |MAS|1.633|0.082|**1.581**|0.066|1.715|0.161|0.195|1.220|0.281|**0.041**|
> > > |PackNet|**1.746**|0.211|–|–|–|–|–|–|–|–|
> > > |AGEM|1.172|0.983|0.668|1.359|0.681|1.493|0.021|0.896|–|–|
> > > |ER-ACE|1.101|0.549|1.092|0.568|1.185|0.682|–|–|–|–|
> > >
> > > The rehearsal-based methods are not competitive with others on Overcooked. Interestingly, AGEM's performance drops when paired with MAPPO or HAPPO, whereas ER-ACE is more consistent across MARL algorithms.

---

### Official Review · Reviewer_6gVS · 2026-03-12

**Soundness:** 2
**Presentation:** 3
**Significance:** 3
**Originality:** 2
**Overall Recommendation:** 5
**Confidence:** 4

**Summary:**

This paper introduces MEAL, a benchmark for continual multi-agent RL based on the Overcooked environment. The authors tackle the main inefficiency of other continual learning benchmarks: the computational inefficiency to handle long-horizon tasks. Additionally, by using a multi-agent environment, the evaluation of common MARL algorithms can be performed. Being built over JaxMARL, the benchmark is highly efficient. The authors also benchmark some CL algorithms and MARL algorithms (IPPO and MAPPO). Various ablations investigate the impact of the number of agents, partial observability, and task sequence length. Although the benchmark is built mainly over one environment, it seems a valuable asset for the incipient CMARL community.

**Compliance With Llm Reviewing Policy:**

Affirmed.

**Final Justification:**

The authors propose the MEAL benchmark, which has the potential to be a valuable asset for the incipient CMARL community.

During the rebuttal phase, the authors effectively addressed my main concerns. As a result, I increased my score to Accept.

I believe that this benchmark can be directly employed by many works that will explore CMARL in the following years (potential to be significant), and that is my main reason for suggesting the acceptance for presentation at the conference.

**Key Questions For Authors:**

Comments:

1. The Figure 3 axis captions and title can be improved so that the reader can understand the metric evaluated in each subfigure without needing to read the caption.
2. The focus on the task-incremental CL paradigm to show the main results leads to better results using multiple heads, but might mislead the direction of the field. Even with the worst results, it would be better to use shared architectures that are conditioned on task ids or implicit task distribution shifts.
3. Line 404-406: Similar to other studies, the authors found that “Contrary to expectation, MAPPO underperforms IPPO”. Can you expand on which changes would be needed in current MARL paradigms to achieve better results than RL algorithms being applied individually for different agents?
4. Additional Discussion regarding what are the properties of CMARL algorithms compared to CL and MARL algorithms would be helpful for readers.
5. In Table 6, can you explain in more detail what “repeated r times” means, parallel environment, or multiple seeds for evaluating the robustness of the algorithms?
6. Another limitation currently in this type of benchmark is the need for dense rewards. For expansion to more general tasks, this might be a limitation, as modeling a dense reward is complex. It would also be interesting to add the sparse reward results to the main paper.
7. The task distribution diversity seems very limited for a continual learning benchmark. The benchmark would be much more robust, adding more task distribution shifts. An example would be to add a permute option in the state/observation over time similar to Juliani, Arthur, and Jordan Ash. "A study of plasticity loss in on-policy deep reinforcement learning." Advances in Neural Information Processing Systems 37 (2024): 113884-113910.

**Limitations:**

Yes

**Strengths And Weaknesses:**

Strengths:

1. The authors proposed a computationally efficient CMARL benchmark based on an Overcooked environment.
2. Results from the authors show that conclusions from long task sequences might differ from those drawn from shorter sequences for continual learning algorithms.
3. The manuscript contains several ablation studies to evaluate the potential of the proposed benchmark and the limitations of current methods.

Weaknesses:

1. The benchmark is focused only on Overcooked (as an expansion from previous work). Another environment is discussed in the Appendix, but it does not seem to be a part of the benchmark.
2. The results are the best and shown mainly for architectures with multiple output heads (one for each task). This assumption might mislead the direction of the field.  We want shared architectures that solve multiple tasks continually in the end.
3. Other benchmarks for continual multi-agent RL with big policies like LLMs are likely to surge. At least discussing this direction might improve the manuscript.

---

> ### Author Rebuttal · Authors · 2026-03-29
>
> **W1 Only Overcooked**. We agree that more environments strengthen the benchmark, as rightfully pointed out by all the reviewers. Note that JaxNav (Appendix M) is already fully a part of MEAL. We have also included two more MARL environments: **MPE SimpleSpread** and **SMAX**. Please refer to our response to **Reviewer bbE3 W1**, where we explain how we use these environments and present the results.
>
> **W2/Q2 Multi-Head Outputs**. Indeed, we agree that the field ought to strive for shared architectures. Our choice of multi-head as the default was to align our evaluation with established CRL benchmarks [1, 2], but since multiple reviewers raised this point, we shall switch to single-head as the default setting to promote this direction of the field. We recently included PackNet in our evaluations, which proved particularly effective in the single-head setting:
>
> |Level|Method|Multi-Head|Single-Head|
> |-----|------|--------|------------|
> |L1|Online EWC|1.612±0.03|0.323±0.06|
> |L1|PackNet|1.746±0.13|0.880±0.01|
> |L2|Online EWC|1.337±0.03|0.407±0.05|
> |L2|PackNet|1.381±0.11|0.561±0.12|
> |L3|Online EWC|1.128±0.11|0.339±0.15|
> |L3|PackNet|1.291±0.04|0.318±0.03|
>
> **W3 LLM**. LLM-based policies are indeed an interesting direction for CMARL, as their in-context learning abilities could offer a useful mechanism for adaptation. In principle, MEAL's JAX-based design is compatible with transformer-based policies. However, MEAL currently has no textual representation that LLMs could leverage. We leave this for future work.
>
> **Q1 Figure 3**. We are not sure how the y-axis labels could be improved, but we did add titles to each subfigure to improve readability.
>
> **Q3 MAPPO vs. IPPO**. We attribute MAPPO's poor performance to its centralized critic, which conditions on joint observations/actions that drift substantially across tasks, yielding noisier targets and stronger cross-task interference. IPPO's independent critics learn simpler task-local value functions that transfer more stably. To improve, centralized methods could make use of mechanisms that decouple cross-agent information sharing from cross-task interference. This is an open problem that MEAL is well-suited to study.
>
> **Q4 CMARL Algorithm Properties**. Our results point to what makes CMARL distinct from our naive MARL+CL combinations. Our strongest baselines work in simple settings but struggle as coordination demands grow (Table 2). MAPPO's centralized critic suffers from task drift (Section 5.5). More agents hurts retention, not just learning (Table 3). This suggests CMARL methods need to compress away layout-specific features in favor of invariant coordination knowledge, encourage diverse agent roles (since homogenized agents risk collective failure when dynamics shift), and selectively reuse learned cooperative patterns rather than replaying everything. We shall include a more detailed discussion in the paper.
>
> **Q5 Repeating Sequences**. In Section 5.6, we repeat training on the same task sequence $r$ consecutive times. This is a known method to study plasticity loss [2]. We mentioned it in the main text, but we will improve the caption of the Table.
>
> **Q6 Dense Rewards**. First, Appendix L already explores the sparse reward setting. We shall move these results to the main paper. Second, although sparse rewards yield lower performance, in principle, MEAL does not require dense rewards. If anything, this performance gap suggests an open challenge. The fact that our naive CL+MARL combinations struggle here is a positive signal: it shows CMARL needs specialized methods. Third, potential *"expansion to more general tasks"* does not diminish the value of the benchmark as it stands.
>
> **Q7 Task diversity**. We appreciate this suggestion and have introduced four additional non-stationarity factors beyond layout changes: (1) randomized **pot size** (1–5 onions), (2) randomized soup **cooking time** (10–30 ticks), (3) **sticky actions** (repeat probability scaling with difficulty: 10–30%), and (4) **slippery tiles** (25% of walkable tiles; slip probability 35–65%). We probe each individually and combined on 100-task Online EWC sequences:
>
> |Dynamics|Level 1|Level 2|Level 3|
> |---|---|---|---|
> |Default|1.612 ± 0.03|1.337 ± 0.03|1.128 ± 0.11|
> |Pot Size|1.588 ± 0.08|1.284 ± 0.04|1.017 ± 0.05|
> |Soup Timer|1.547 ± 0.11|1.269 ± 0.03|1.060 ± 0.05|
> |Sticky Actions|1.475 ± 0.03|1.177 ± 0.03|0.839 ± 0.11|
> |Slippery Tiles|1.365 ± 0.03|1.161 ± 0.02|0.832 ± 0.05|
> |**Combined**|1.124 ± 0.07|0.948 ± 0.03|0.666 ± 0.04|
>
> Individually, each source of non-stationarity mildly reduces performance. When combined, their effects accumulate and substantially increase task difficulty.
>
> [1] Wołczyk, Maciej, et al. "Continual World: A Robotic Benchmark for Continual Reinforcement Learning." Advances in Neural Information Processing Systems 34 (2021).
>
> [2] Tomilin, Tristan, et al. "COOM: A Game Benchmark for Continual Reinforcement Learning." Advances in Neural Information Processing Systems 36 (2023).

---

> > ### Author Rebuttal · Reviewer_6gVS · 2026-04-01
> >
> > Thank you for addressing my comments.
> >
> > I do believe that addressing comments Q2 (single-head as default) and Q7 (additional non-stationarity) in the manuscript will increase its impact substantially.
> >
> > Still, I want to follow-up into two comments that are important to be discussed in the manuscript.
> >
> > W3. I understand that big policies are not the main focus of the paper, but a discussion on how these large models can be applied, or what the challenges are, in the CMARL field would be helpful. As this is a personal preference from the reviewer, the authors may add this or not to the final version.
> >
> > Q3 and Q4. It is crucial that the discussion made in these points is incorporated into the paper. For example, even Section 5.5 discusses partial observability, but it is hard to get any additional insights regarding the centralized critic aspects. In this case, discussing how the task drift affects the critic seems very important.
> >
> > I will increase my current score to accept, given that most of my concerns were addressed effectively.

---

> > > ### Author Response · Authors · 2026-04-05
> > >
> > > Thank you for the positive reassessment and constructive suggestions.
> > >
> > > We will briefly draft our ideas for the large models (W3) discussion part. The most obvious challenges (although not uniquely CL-related) with environments such as in our work are the **evaluation time** and **inference cost**. MEAL can simulate up to 1M steps per second because policy inference is fast. With large models, these performance gains are lost, and evaluation at the same granularity would not be practical, especially with a growing number of agents and tasks. Generally, large policies are used hierarchically in such settings. They periodically devise a strategy/plan for an agent given the circumstances. Then a low-level controller carries out that strategy/plan.
> > >
> > > More related to CMARL, one clear challenge is the **context length**. If adaptation is done in-context, each agent must maintain a representation of how previous tasks worked and how to cooperate with others across the full task sequence. Over 100 tasks, this can become infeasible. It is also not trivial how coordination should be represented in text. A related challenge is **communication overhead**. A practical way to make LLM-based agents work in cooperative settings would be to allow them to explicitly communicate or broadcast their plans. Our decentralized RL policy baselines learn to infer what other agents intend without any explicit communication. While this simplifies the problem considerably, it is an open challenge to make such communication work across many tasks. There is also the question of how much time and resources are spent on it. That said, LLM-based approaches could be particularly interesting for MEAL's partner adaptation setting (Appendix K), where reasoning about a new partner's behavior from observation is closer to high-level inference than frame-by-frame control.

---

### Official Review · Reviewer_zp1k · 2026-03-13

**Soundness:** 3
**Presentation:** 3
**Significance:** 3
**Originality:** 3
**Overall Recommendation:** 4
**Confidence:** 4

**Summary:**

MEAL introduces a JAX-based benchmark for continual multi-agent reinforcement learning (CMARL), built on the Overcooked environment. The contributions are: (1) a procedural generator across three difficulty levels, (2) GPU acceleration of the overcooked environment, and (3) evaluation of some CL methods (EWC, Online EWC, L2, MAS, AGEM) and IPPO as a baseline, mainly in the task incremental continual learning setting.

**Compliance With Llm Reviewing Policy:**

Affirmed.

**Final Justification:**

My main concerns have been addressed , I recommend the authors to take them into account to improve the paper.

**Key Questions For Authors:**

I have inserted my questions above since they are related to each category.

**Limitations:**

The authors acknowledge key limitations, including the benchmark being limited to Overcooked. So this is addressed.

They do not address any potential negative societal impact (i.e., the required ICML disclaimer seems to be missing at the end).

**Strengths And Weaknesses:**

Soundness

The benchmark is technically sound and has a range of nice properties, such as procedural generation. The fact that it is written in JAX, and thus has GPU acceleration, is almost crucial nowadays, and is the right design choice. The authors also show that because JAX enables more episodes, certain effects of continual learning only show up after many tasks/episodes, which is an important insight and only possible with enough throughput.
The main weakness to me here is the limited focus on Overcooked. Although I acknowledge that something like SMAC was also published as a single environment, it contained multiple game modes to test algorithms on, and was a novel benchmark in terms of benchmark mechanics for MARL research. Overcooked, however, already exists for a while as a (non-GPU accelerated, non continual learning) benchmark (i.e., Joel Leibo's MeltingPot). To me, the paper would have been much stronger if other games with different mechanics were also included in the benchmark, all crafted with the same rigor as Overcooked. The authors are aware of this by acknowledging this in the paper's limitations and argue they choose depth over breadth, but to me this remains a major shortcoming.
Some other questions:
You say: "In our experiments, we adopt the task-incremental CL paradigm, in which the task identity is known during both training and evaluation." —> While I appreciate the transparency about this choice, the exclusive focus on task incremental learning narrows the benchmark's usefulness. Domain-incremental or class-incremental settings, where the agent must infer or adapt to the task without an explicit identifier, arguably better reflect realistic deployment scenarios. I see in the ablation, you test without task identifiers, and it seems to not matter too much. Why not leave out task identifiers entirely then? In the same line: the use of multi-head outputs (shown to be the most critical architectural component in the ablation) is tightly coupled to this paradigm. Would the benchmark still be informative under settings where automatic output head selection is unavailable?
In 'Experiments', you say you opt for 100 tasks mainly, and often show 20 tasks in your plots (indicated as for clarity). In Figure 3 (bottom), the forward transfer comparison against IPPO uses only 10 Level 2 tasks, though? Given your claim that certain effects only appear at longer sequences (up to 100 tasks), it would strengthen the paper to show forward transfer analysis on longer sequences as well. Maybe I am missing something here.
For the curriculum learning: the result nearly doubles performance on Level 3 but the difference on Level 2 is not statistically significant given the variance. Is the curriculum finding robust, or could it be an artifact of the particular task samples?
Were CL method hyperparameters (e.g., lambda for EWC/MAS/L2, memory size for AGEM) tuned per difficulty level or kept fixed across all levels? The appendix suggests a single fixed value per method, but optimal regularization strength likely differs across difficulty levels. This matters for the fairness of the comparison, since a method could appear to fail on harder levels simply because its hyperparameter was tuned for easier ones.
Additionally, could the normalization scheme (dividing by a single-agent optimal bound) distort multi-agent comparisons? Scores above 1.0 reflect cooperation gains, but the bound itself is loose (it ignores parallelism). This means two methods with different cooperation levels might appear more similar than they are.

Presentation

In general, the paper is well-presented, although I found some sections unclear.
With regards to wall-clock time, do you report this for all tested algorithms, or simply when running the environment steps in isolation?
You say "During training, we evaluate the policy after every 100 updates" —> every 100 update steps? This seems quite frequent. Are these gradient updates or episodes? And then, "by running 10 evaluation episodes on all tasks in the sequence": does this mean 10 episodes for each task that was already seen so far? This was unclear to me.
Then you say: "The results are displayed with 95%…": are displayed where exactly? Is this sentence talking about all results?
Is L2 simply weight decay applied to parameter changes (penalizing drift from previous task parameters)? You refer to Kirkpatrick et al. (2017) when mentioning L2, but that paper is mainly about EWC, so the reference is slightly misleading. It would help to clarify this, especially since there is also a distinct continual learning method called L2-init from Kumar et al. (2023) ("Maintaining plasticity in continual learning via regenerative regularization") that readers might confuse it with.

Significance

The paper addresses a real computational bottleneck that genuinely constrains CRL research to short sequences, and finds that experimental conclusions change substantially with sequence length, which is a non-trivial insight for the field beyond just the benchmark. CMARL as a setting is underexplored and the paper makes a good case for why it introduces new challenges beyond single-agent CRL. However, as the main weakness, which is related to the main weakness in 'soundness': I would have liked to see more environments. If the field wants to test new CMARL algorithms, we need more than just Overcooked, and it would make the contribution stronger when researchers can simply refer to a range of environments that are centrally gathered in your repository, instead of relying on this single game (which might not be representative of other game mechanics or testing scenarios: how confident can we be that findings generalize beyond soup delivery?). In the appendix the authors mention jaxnav too, but it is not integrated in the main experiments. The CMARL community is small, and therefore it would be much more useful to have this presented as a default benchmark setting (for which I believe the main elements are here already), but again conditioned on it having more settings/environments. Why not port more MeltingPot games?

Originality

The diverse partner adaptation setting is an interesting and underexplored problem. However, at the same time, it feels underdeveloped for something listed as a contribution (its evaluation is one paragraph in the appendix, it seems?).
The procedural generation pipeline with validation, the N-agent extension, the partial observability variant, and the designated-roles experiment collectively represent meaningful engineering work, but the conceptual novelty is limited. The game itself, Overcooked, is not new and was already a benchmark. The originality lies primarily in the adaptation for a specific field, CMARL.

---

> ### Author Rebuttal · Authors · 2026-03-30
>
> We appreciate the very thorough and thoughtful review!
>
> **Only Overcooked**. We would slightly push back on a few points. First, single game/domain benchmarks are not that uncommon in both MARL (SMAC, Hanabi, Google Research Football) and CRL (COOM & CRLMaze: ViZDoom, Jelly Bean World, Continual World & Continual Bench: robotic arm). SMAC's "game modes" are rather variations in unit compositions, initial conditions, and terrain. MEAL's layout generation and difficulty scaling are not that distant from it. Second, in CL, the complexity lies more in the task *sequence*, not only the base RL/MARL environment. Even a simple domain can impose a challenge for CL, especially when scaling up to 100 tasks. Third, Overcooked was never part of MeltingPot. MeltingPot contains Sequential Social Dilemma (SSD) games. We think this falls out of our scope and could deserve a paper of its own.
>
> That said, we agree that more environments are valuable. Beyond JaxNav (which we will move to the main paper), we now include **MPE SimpleSpread** and **SMAX**. Due to the character limitation of our response and the many good comments you've made, which we wish to still respond to, please refer to our response to **Reviewer 6gVS W1**, where we introduce these environments and present the results.
>
> **Task-Incremental Only**. We agree that shared architectures are the end goal. We shall adopt single-head as the default. Please see our response to Reviewer 6gVS (W2) for our reasoning, along with single-head results of Online EWC and PackNet.
>
> **Sequence Length**. The exact 100-task results are already reported numerically in Table 2, including forward transfer. The 10- and 20-task figures mainly serve the purpose of helping readers understand how the metrics work by *visualizing* how forgetting/transfer manifests over time. 100-task plots are not legible at this resolution. The plotting code is included in the repository for users who want to inspect specific task windows.
>
> **Curriculum Robustness**. Each seed we run generates different layouts, so we wouldn't attribute this to sampling artifacts. Curriculum helps Level 3 because larger grids require agents to memorize longer precise action sequences, making agents more prone to overfitting. Pre-training on 5 Level 1 + 5 Level 2 tasks guides exploration on Level 3 more effectively than 10 other Level 3 layouts. On Level 2, this issue hasn't emerged. Agents can learn better from scratch.
>
> **Hyperparameters**. We tuned hyperparameters on Level 1 and kept them fixed across levels, since the alternative values (different λ for regularization methods, different memory/batch sizes for AGEM) we tested did not improve performance.
>
> **Normalized Soup Metric**. The bound is intentionally conservative. It computes the optimal single-agent cycle time assuming no parallelism. This means scores above 1.0 indicate genuine cooperation gains. We don't see why the bound being loose would distort comparisons. It is applied uniformly to all methods on the same layout, so relative differences are preserved. A tighter bound would simply rescale all scores without changing rankings.
>
> **Wall-Clock Times**. Thank you for pointing this out. We will add per-method timing for 100-task Level 1 sequences, including evaluation on a single H100 GPU.
> MARL algorithms (with FT): IPPO 4h 32m, MAPPO 5h 24m, HAPPO 7h 49m. HAPPO is slower due to sequential per-agent updates. CL methods (all IPPO-based): FT 4h 32m, EWC 4h 36m, L2 4h 41m, MAS 4h 47m. CL overhead is negligible.
>
> **Clarifications**. (1) Every 100 PPO update steps (each consisting of 8 epochs × 16 minibatches = 128 gradient steps). (2) Yes, all tasks in the sequence are evaluated at every checkpoint, including unseen future tasks. This is intentional as it enables visualizing forward transfer. (3) "95% confidence intervals" applies to all the intervals that we do report, in both tables and figures. (4) L2 is simply L2 regularization on parameter drift from the previous task's parameters, a naive baseline, not an established CL method. We cite the EWC paper because it compares against it. We will clarify these points in the paper.
>
> **Partner Adaptation**. The contribution here is the *"infrastructure"*: task sequences with pre-trained diverse partners (BRDiv populations, planning-based agents, hardcoded strategies) ready for evaluation. We show to what extent performance is lost using the naive fine-tuning approach. A simple L2 regularization can slightly mitigate this.
>
> **Novelty**. We deliberately based MEAL foremost on Overcooked because it is widely adopted and researchers are familiar with it. This pattern is well-established: SMACv2 kept StarCraft but fixed stochasticity, MeltingPot 2.0 replaced scenarios but kept the platform, Gymnasium cleaned up OpenAI Gym's API. The evaluation settings are not meant to be novel in themselves. What matters is that they are meaningful and readily available in the benchmark.

---

### Decision · Program_Chairs · 2026-04-30

**Decision:**

Accept (regular)

**Comment:**

The recommendation to accept is based on this work providing a
benchmark with solid design decisions in an area that needs it: CMARL.
Use of JAX and GPUs allows Overcooked to go up to 100 procedurely
generated tasks.  There is an interesting comparison of some baseline
methods.  As with any benchmark, there are many design decisions and
the wish list of additions is long.  Through the rebuttal process, the
authors added some of the requests, which improved the scores.